# Cross-scale drivers of soil fungal diversity in fragmented forests of southwestern China
Yawen Lu [1], Jing Han[1], Shilu Zheng [2] & Ying Chen [3] ✉

While numerous studies have explored the factors influencing soil fungal diversity, few compared the impact of environmental variables across multiple spatial scales. Based on 30 plots spanning a fragmentation gradient in tropical forests of Xishuangbanna, southwestern China, we predicted the diversity and composition of nine soil fungal functional guilds with environmental factors across local-, patch-, and landscape-scales. The results showed that local tree composition was the most dominant factor associating with almost all the detected fungal functional guilds, explaining an average of 45.72% relative contribution. Although patch and landscape factors contributed less, the mean patch area at the landscape scale still showed a significant negative association with the diversity of fungal functional guilds. Some fungal functional guilds showed contrasting patterns to environmental factors. Specifically, the richness and abundance of ectomycorrhizal fungi exhibited opposite trends to forest types relative to other guilds, whereas the community composition of soil/unspecified saprotrophic fungi showed contrasting responses to tree composition compared with other guilds. Our results emphasize that local environments (i.e., soil and tree) remain the main factor affecting soil fungal diversity and composition in tropical forests. Therefore, maintaining diverse forest types and compositional heterogeneity of tree species can help future fungal conservation efforts.

Soil-inhabiting fungi are among the largest contributors to global biomass and play essential roles in nutrient cycling[1], species co-existence[2,3], and productivity of terrestrial ecosystems[4]. Soil saprotrophic fungi, for example, can decompose organic matter and plant litter, contributing to soil carbon accumulation and supporting plant growth[5]. Beneficial symbiotic fungi (e.g., arbuscular mycorrhizal and ectomycorrhizal fungi) can promote plant growth by increasing the availability of host plant roots to nitrogen (N) and phosphorus (P) in the soil[2]. In contrast, pathogenic fungi can negatively impact plant growth but maintain plant diversity by suppressing the dense growth of conspecific host plants[6]. As fungi are classified into multiple functional types and differ in their nutritional habits, the factors shaping their diversity and composition may vary substantially among guilds[7]. For example, due to the host specificity of ectomycorrhizal fungi, higher proportions of ectomycorrhizal plants in communities often correspond to greater soil ectomycorrhizal fungal richness[2,8]. However, most studies only explore the factors influencing soil fungal diversity within a specific spatial scale[9,10], with few comparing the impact of environmental factors across multiple spatial scales[11], resulting in underestimation of their spatial structures.

At local scale, the biotic and abiotic environmental factors usually interact with soil fungi directly, driving fungal ecological processes and diversity changes[12]. For example, local vegetation characteristics (including plant richness, composition, and biomass) and soil properties (including pH, carbon content, and elemental stoichiometry) are found to be key drivers of soil fungal diversity[9,13,14]. A global meta-analysis reveals that higher soil pH is positively correlated with soil fungal richness and Shannon diversity[15], while some regional studies have reported that acidic soils may also promote fungal biomass or growth, particularly in abandoned land systems[16,17]. Additionally, elevated soil N levels are linked to an increase in soil pathogenic fungi and a decrease in mycorrhizal fungal diversity[14,18]. Higher plant diversity has been found to increase overall soil fungal richness[8], but it decreases the abundance of specific functional guilds (e.g., pathogenic fungi)[19,20]. Given their direct influence on soil fungi and mediation of belowground trophic processes, local-scale environmental factors

[1]College of Chemistry and Life Science, Suzhou University of Science and Technology, Suzhou, 215009, China. [2]MOE Key Laboratory for Biodiversity Science and Ecological Engineering, Institute of Biodiversity Science, School of Life Sciences, Fudan University, Shanghai, 200438, China. [3]Tiantong National Station for Forest Ecosystem Research, Institute of Eco-Chongming (IEC), The Shanghai Key Lab for Urban Ecological Processes and Eco-Restoration, School of Ecological and Environmental Sciences, East China Normal University, Shanghai, 200241, China. ✉e-mail: yingchen@des.ecnu.edu.cn

are widely considered in microbial diversity predictions[13]. However, which factor is the strongest contributor is still a matter of debate[8], and a more comprehensive comparison among local-scale environmental factors is needed. In addition, although many studies have primarily focused on alpha diversity metrics such as species richness of soil fungi, changes in community composition (beta diversity) can also reveal shifts in species identities and turnover[21,22]. These patterns offer equally important insights into biodiversity responses and therefore merit comprehensive and systematic investigation.

Habitat fragmentation caused by anthropogenic land use change is affecting soil fungal diversity at larger spatial scales[10,11]. At patch scale, fragmentation reduces patch area and increases edge effects (i.e., changes in community structures near habitat boundaries), but its specific impact on soil fungal diversity remains poorly understood[23,24]. Studies generally consider that habitat fragmentation can reduce fungal diversity[23,25]. For example, in Hawaii's lava-fragmented forests, belowground root-associated fungal diversity increases with patch area[25]. Other studies have shown that the effect of patch-scale environmental factors vary with fungal functional guilds[10]. The species richness of wood saprotrophic fungi decreases rapidly with loss of patch area[26]. The community composition of arbuscular mycorrhizal and ectomycorrhizal fungi has been shown to vary with edge distance[27]. Some research, however, suggests that the impact of environmental factors at patch scale is weak[23,25]. For instance, the patterns for pathogenic fungi in relation to patch area and edge distance are not clear[23]. These mixed results may partially reflect taxon-specific responses to fragmentation, arising from differences in host specificity, dispersal abilities, and habitat affiliations[10]. Moreover, as some fungal guilds have the ability to disperse across larger scales (up to 10 km)[28], patch-scale factors may be insufficient to predict soil fungal diversity.

To address this, Grilli et al. proposed a landscape-scale concept to predict fungal diversity by considering overall landscape conditions, that is, the patch composition and distribution within a specific spatial radius[23]. As the alternation of a habitat's spatial configuration after fragmentation, it may reduce the potential for soil fungi to colonize and persist in the remaining habitat patches[29]. Some recent studies have detected that the landscape-scale factors of patch or edge density, and landscape amount at the spatial radius of 500- to 2000-m have strong effects on soil fungal diversity[11,30]. However, most current studies do not distinguish the effects between patch- and landscape-scale environmental factors when detecting the impact of fragmentation on biodiversity, which may obscure their respective contributions[31,32]. Therefore, comparing the significance and relative contributions between patch- and landscape-scale environmental factors are necessary for better prediction of soil fungal diversity under habitat fragmentation.

Environmental factors at different spatial scales can also interact with each other. For example, local environmental conditions can be affected by habitat fragmentation, such as variations in soil nutrient availability at fragment edges due to exposure to surrounding matrix habitats[24,27]. Furthermore, the forest type also shapes plant composition and soil conditions, impacting soil fungal diversity[33]. Specifically, lowland forests (600–900 m elevation) are warmer and characterized by tall megaphanerophytes (often > 30 m) and large woody lianas[34]. Montane forests (above 900 m elevation), with cooler temperatures, are dominated by shorter trees with leathery, microphyllous leaves[34]. The diverse vegetation composition may result in more fungal diversity in lowland and montane forests[33]. In contrast, limestone forests are species-poor, dominated by calciphilous species, and experience more variable microclimates, implying the presence of functionally distinct taxa of fungi[32]. Moreover, soils in limestone forests are calcium-rich and near-neutral (pH ~6.5), while lowland and montane forests develop on more acidic soils (pH 4.5–5.5)[34]. These sharp ecological contrasts offer a unique setting for exploring cross-scale drivers of fungal diversity[32,33]. However, the relationship and relative strength of these factors in determining soil fungal diversity and community composition remain poorly understood[10,31]. Accurate predictions that integrate local-, patch- and landscape-scale

environmental factors are needed to better understand the diversity of various functional guilds of soil fungi.

By collecting soil samples in 30 plots across 17 forest fragments and three forest types in Xishuangbanna, southern China, we calculated the richness, Shannon diversity, community composition and abundance of soil fungi with different functional guilds to comprehensively compare the association and relative contribution of local-, patch-, and landscape-scale environmental factors. Local-scale factors include soil properties, tree species diversity of richness, Shannon diversity and community composition, elevation and slope. Patch-scale factors include patch size and edge distance of the sampling plot. Landscape-scale factors include mean of patch area (mean area), edge density, patch density, and patch richness within a 2 km spatial radius of the sampling plot. Specifically, we propose the following hypotheses: (i) Local-scale vegetation characteristics and soil properties will be the primary drivers of soil fungal diversity in fragmented forests. (ii) Patch- and landscape-scale factors will also exert significant effects on soil fungal diversity, providing complementary contributions to the variation explained beyond those of local factors. (iii) Different soil fungal functional guilds will exhibit distinct responses to environmental factors across spatial scales, reflecting their distinct ecological strategies and environmental dependencies. Here, we provide evidence that local tree composition predominantly drives soil fungal functional guild diversity, while patch- and landscape-scale factors exert weaker yet significant influences. Moreover, different fungal functional guilds exhibit distinct responses to environmental factors across spatial scales.

## Results
### Taxonomic and functional assignment of fungi
Among the 8288 fungal OTUs detected, the 30 soil samples yielded an average of $39,939.9 \pm 2163.5$ (mean ± s.e.) sequences before rarefaction, ranging from 23,991 to 87,999 sequences per sample. After rarefaction and fungal functional guild assignment, a total of 364,654 sequences representing 2385 OTUs were assigned to the nine fungal functional guilds, accounting for 30.4% of all fungal sequences and 28.8% of all OTUs detected (Fig.1; Supplementary Fig. 1 and Supplementary Table 1). Ectomycorrhizal fungi contributed the largest proportion of sequences (144,777) among the nine guilds, of which 89.7% belonged to the phylum Basidiomycota (Fig. 1; Supplementary Fig. 1b and Supplementary Table 1). Soil saprotrophs were the second most abundant guild, comprising 90,988 sequences and exhibiting the highest OTU richness (518 OTUs) (Fig. 1 and Supplementary Table 1). This guild was taxonomically diverse, including phyla of Ascomycota (26.4%), Basidiomycota (44.2%), Mortierellomycota (13.5%), and Mucoromycota (15.6%) (Supplementary Fig. 1b). Foliar endophytes had the fewest sequences (2779) and OTUs (39 OTUs) among the nine guilds, all of which belonged to phylum Ascomycota.

### Correlation between cross-scale drivers
We found that patch- and landscape-scale factors were weakly correlated with local-scale factors, with the exception of a negative correlation between edge density and elevation ($r = -0.74$; $p < 0.001$) (Supplementary Fig. 2). This indicated that that fragmentation was not significantly associated with local tree species diversity, composition, and soil properties in the region. Local-scale factors, including soil properties (i.e., soil PC1) ($p < 0.001$), tree composition ($p < 0.001$), tree richness ($p < 0.001$), and elevation ($p = 0.045$) were mainly shaped by forest type. However, patch- and landscape-scale factors did not differ significantly among forest types (Supplementary Fig. 3).

### Cross-scale drivers on soil fungal diversity
Forest type significantly affected the diversity of nine fungal functional guilds. The richness and abundance of most guilds were significantly higher in limestone forests, followed by lowland and mountain forests, except for ectomycorrhizal fungi, which exhibited an opposite pattern (Fig. 2a, d). For Shannon diversity, the patterns of different fungal functional guilds were generally similar. Limestone forests showed significantly higher Shannon

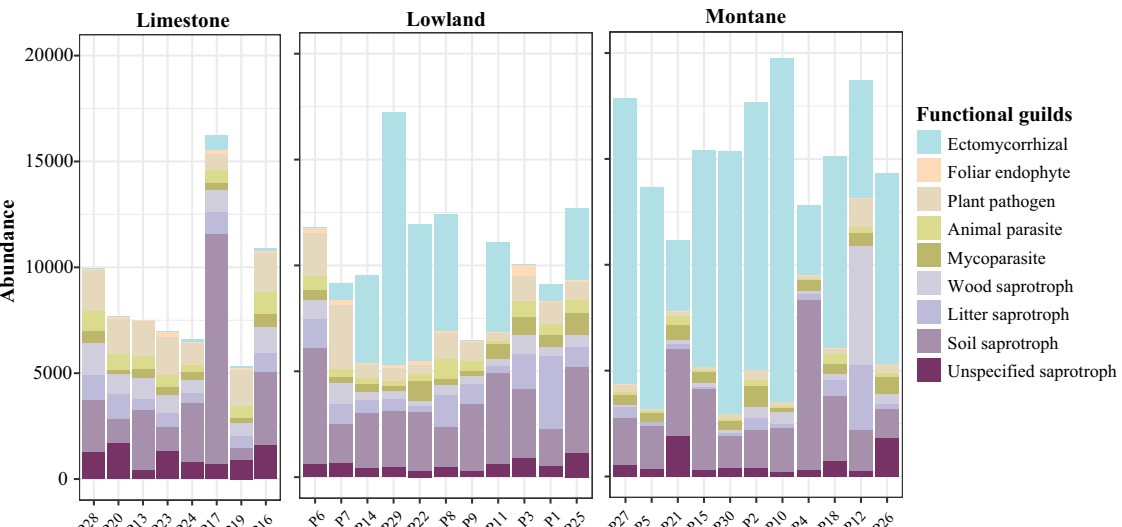

**Fig. 1 | Soil fungal abundance across three forest types.** Rarefied sequence counts of soil fungi assigned to the represented functional guilds across 30 sampling plots in limestone, lowland, and montane forests.

diversity than lowland forests (for ectomycorrhizal fungi and mycoparasites) and montane forests (for all fungal functional guilds) (Fig. 2b). The community composition, represented by the first axis of a PCoA (PCoA1) based on Bray-Curtis dissimilarity, of different fungal functional guilds also showed the similar pattern, with limestone forests having the greatest compositional separation, followed by lowland and mountain forests, except that soil and unspecified saprotroph showed the contrast pattern (Fig. 2c).

Spearman rank correlation showed soil PC1 and tree composition were significantly correlated with most fungal functional guilds, with the four diversity indices positively correlated with soil PC1 (except for richness of ectomycorrhizal fungi, community composition of soil and unspecified saprotroph, abundance of ectomycorrhizal fungi and mycoparasite) (Fig. 3; Supplementary Fig. 4 and Supplementary Table 2), but negatively correlated with tree composition (except for richness of ectomycorrhizal fungi, community composition of soil and unspecified saprotroph, abundance of ectomycorrhizal fungi, mycoparasite and soil saprotroph) (Fig. 3; Supplementary Fig. 5; Supplementary Table 2 and 3). The correlations between fungal diversity and patch- or landscape-scale factors were significantly weaker than those with local-scale factors ($t = 10.34$; $p < 0.001$ comparing with patch-scale factors and $t = 5.21$; $p < 0.001$ with landscape-scale factors). Specifically, mean area at landscape scale showed significant positive correlations with richness and abundance of ectomycorrhizal fungi, but negative correlations with other fungal functional guilds (not all significant) (Fig. 3 and Supplementary Table 2).

Multivariate linear mixed effect models (patches and forest types as random effects) verified significant effects of tree composition (PCoA1) on the diversity of nine fungal functional guilds. The community composition (PCoA1) of all fungal functional guilds was significantly affected by tree composition. Specifically, saprotrophs (standardized regression coefficient $[\beta] = 0.425$, $p = 0.017$) and unspecified saprotrophs ($\beta = 0.876$, $p < 0.001$) showed positive relationships with tree community composition, indicating that their compositional variation followed the same gradient as tree communities (Fig. 4). In contrast, other fungal functional guilds displayed negative relationships, meaning their compositional variation occurred in the opposite direction along the tree compositional gradient (Fig. 4). Tree composition also showed a positive relationship with richness of ectomycorrhizal fungi ($\beta = 0.565$, $p = 0.005$) and a negative relationship with that of other fungal functional guilds (except soil and unspecified saprotroph were not significant) (Supplementary Fig. 6). Fungal abundance also showed a similar result, with abundance of ectomycorrhizal fungi ($\beta = 0.569$, $p = 0.005$) increasing significantly, and other fungal functional guilds decreasing with tree composition

(Supplementary Fig. 7). Shannon diversity was negatively associated with tree composition, which was significant in foliar endophyte ($\beta = -0.550$, $p = 0.027$), plant pathogen ($\beta = -0.408$, $p = 0.023$), animal parasite ($\beta = -0.498$, $p = 0.015$), mycoparasite ($\beta = -0.600$, $p = 0.002$), and wood saprotroph ($\beta = -0.747$, $p = 0.001$) (Supplementary Fig. 8). However, the effects of soil PC1 on fungal functional guilds were not significant in all diversity indices based on multivariate linear mixed effect models, indicating the weak effect or variation of soil properties within the forest type and patch (Fig. 4 and Supplementary Figs. 6–8).

Patch- and landscape-scale factors had significant associations with the diversity of partial fungal functional guilds in the multivariate linear mixed effect models. At landscape scale, higher mean area was negatively associated with community composition of foliar endophyte ($\beta = -0.421$, $p = 0.042$), plant pathogens ($\beta = -0.224$, $p = 0.029$), animal parasites ($\beta = -0.206$, $p = 0.028$) (Fig. 4), richness of foliar endophyte ($\beta = -0.465$, $p = 0.050$), litter saprotroph ($\beta = -0.344$, $p = 0.040$) (Supplementary Fig. 6), abundance of foliar endophyte ($\beta = -0.677$, $p = 0.022$), animal parasite ($\beta = -0.353$, $p = 0.039$) significantly (Supplementary Fig. 7), but positively associated with abundance of mycoparasite ($\beta = 0.514$, $p = 0.048$) (Supplementary Fig. 7). Additionally, patch richness at landscape scale was negatively associated with composition of unspecified saprotroph ($\beta = -0.381$, $p = 0.003$), but increased abundance of unspecified saprotroph ($\beta = 0.560$, $p = 0.028$) and Shannon diversity of plant pathogen ($\beta = 0.844$, $p < 0.001$). At patch scale, higher patch size was negatively associated with abundance of litter saprotroph significantly ($\beta = -0.712$, $p = 0.014$) (Supplementary Fig. 8).

Through model selection, tree composition (or its combinations) emerged as the best predictor for 28 out of 36 diversity combinations across the nine fungal functional guilds (Supplementary Table 4).

## Relative contribution of cross-scale drivers

In the multivariate linear mixed effect models, tree composition was the strongest predictor, explaining $45.72 \pm 3.49\%$ (mean ± s.e.) of the variance across the four diversity indices of the nine functional guilds, followed by soil PC1 ($14.75 \pm 1.52\%$) and tree richness ($8.11 \pm 1.88\%$). Mean area at the landscape-scale had the highest explanation among the factors associated with fragmentation, reaching a relative contribution of $7.86 \pm 2.18\%$ (Figs. 4 and 5a).

Across scales, the total relative contribution of local-scale factors was significantly higher than that of patch- and landscape-scale factors; however, no significant difference was observed between the contributions of patch- and landscape-scale factors (Figs. 4 and 5b).

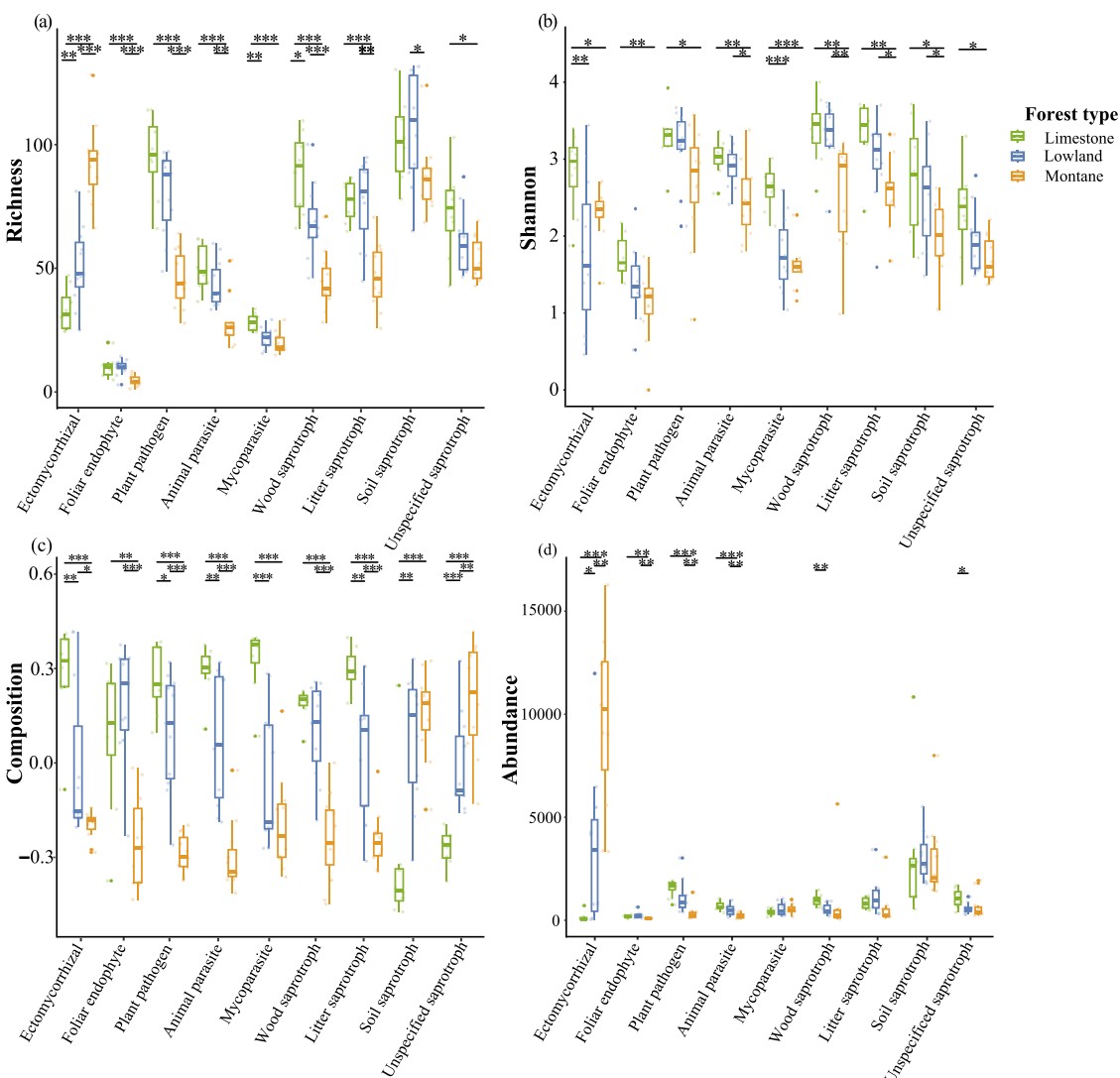

**Fig. 2 | Variation in soil fungal diversity of different functional guilds under three forest types.** Variation in the richness (**a**), Shannon diversity (**b**), community composition (**c**), and abundance (**d**) of each fungal functional guild across limestone, lowland, and montane forests ($n = 30$ biologically independent samples). The significance between forest types for each fungal functional guild was tested by $p$-value with TukeyHSD test, as *** $p \leq 0.001$; ** $p \leq 0.01$; * $p \leq 0.05$.

## Discussion

In the context of global habitat fragmentation, our study comprehensively evaluated the effects of local-, patch- and landscape-scale environmental factors on the diversity of soil fungal functional guilds in tropical forests of Yunnan, China. We found that local tree composition was the dominant driver of the four diversity indices across the nine functional guilds, explaining on average 45.72% of the variation. While patch- and landscape-scale factors played relatively smaller roles, the landscape factor of mean patch area still significantly reduced diversity in most fungal functional guilds. Additionally, different fungal guilds showed contrasting responses: ectomycorrhizal fungi were more diverse and abundant in limestone forests in contrast to other guilds, and soil/unspecified saprotrophic fungi exhibited opposite compositional patterns in relation to tree composition. These findings highlight the distinct responses of fungal functional guilds and diversity indices to fragmentation, shaped by the pronounced ecological heterogeneity of tropical forests.

The significant influence of tree composition in predicting the diversity of fungal functional guilds is particularly noteworthy. However, previous studies have often focused on the alpha diversity metrics of plant diversity, such as species richness, Shannon diversity, and evenness, when assessing the impact on soil fungal diversity[8,33]. This typically underestimates the effect of beta diversity[21], i.e., changes in plant community composition. The association between tree composition and soil fungal diversity is consistent with earlier experiments[9,18,22]. For instance, Makiola et al. demonstrated that plant community dissimilarity is a dominant factor in explaining the diversity of fungal pathogens in soils[9]. Additionally, we found that tree species richness is positively associated with the diversity of several fungal functional guilds, but the effect is weaker than tree composition. Other studies have also observed the similar weak effects of plant richness[1,20]. This might be attributed to the inherently high species richness in tropical forests, where diversity gradients are less pronounced[32,35]. Furthermore, the complexity of natural environments, compared to controlled plant species richness experiments, may obscure observable effects[20]. Conversely, the impact of tree composition may better represent the actual forest structure and pronounce diversity dissimilarities[9]. As Ferlian et al. found, the identity and composition of neighboring trees have a greater influence on mycorrhizal fungal diversity than tree species richness[36].

Local soil environments can mediate the survival and growth of soil fungi greatly[37]. We found that the diversity indices of the detected fungal functional guilds increase primarily under conditions of higher pH and nutrient content, such as elevated levels of N, P, and Ca. Similar findings have been observed in aboveground plant-associated fungi[15,32], as fungi depend on these nutrients for growth and reproduction[13,14]. The soil pH value in our fields ranged from 4 to 7, and fungal diversity generally

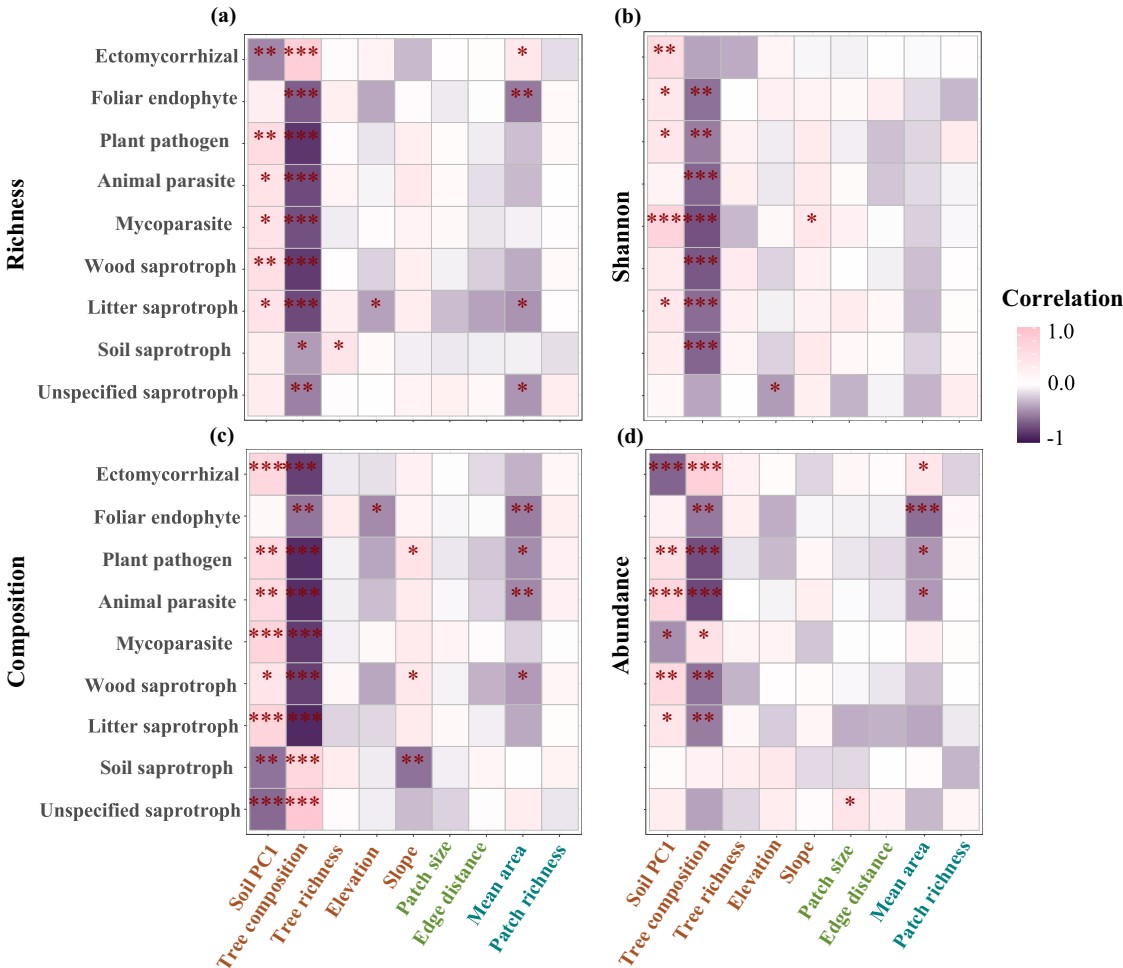

**Fig. 3 | Correlation between fungal functional diversity and multi-scale environmental factors.** Spearman rank correlation matrix between diversity indices of richness (**a**), Shannon diversity (**b**), community composition (**c**), and abundance (**d**) of different fungal functional guilds and environmental factors at local-, patch- and landscape-scales ($n = 30$ biologically independent samples). Local factors include soil PC1, tree composition, tree richness, elevation and slope. Patch factors include $\log_{10}$ transformed patch size and edge distance. Landscape factors include $\log_{10}$ transformed mean of patch area (mean area), and patch richness. The significance was tested by $p$-value as *** $p \le 0.001$; ** $p \le 0.01$; * $p \le 0.05$. Pink shading represents positive correlations, and purple shading represents negative correlations.

increased along this gradient. This suggests that while some fungi can tolerate more acidic soils, diversity tends to be lower under such conditions[15]. However, after controlling for forest types, the significance of soil properties disappeared, indicating that the changes in soil properties were mainly due to different forest types, with minimal variation within each forest type. Since lowland and montane forests are mainly the acidic soil (pH of 4.5–5.5), while limestone forest has the soil pH value around 6.5[32,34].

In comparison, the relationship between the diversity of soil fungal functional guilds and patch or landscape factors was weaker than that with local factors, which is consistent with patterns in regional and global studies[23,25,38]. For example, a comprehensive analysis of soil fungi in northern Europe by Tedersoo et al. revealed that soil properties predominantly explain the diversity and composition of soil fungi, although fragmentation still plays a significant role[8]. Similarly, Newbound et al. observed that soil properties, rather than landscape context, influence woodland fungal communities along an urban-rural gradient[38]. Several reasons may account for this weaker relationship. First, the dispersal ability of soil fungi may be lower than expected, resulting in the minimal impact of large-scale patch structures on fungal diversity[39]. Some studies considered that certain fungal guilds can disperse over distances of only a few dozen meters[40]. For example, fungi with larger spores, such as mycorrhizal fungi and some saprotrophic fungi, tend to reduce dispersal ability and be localized with host plant species[13,40,41]. Second, the spatial scale effects of landscape factors may not be

fully considered[42]. Researches detected that fungal diversity pattern usually changes with spatial scales[11,31]. For instance, Su et al. identified 400 and 600 m as the optimal landscape scales for predicting soil fungal diversity in the agro-pastoral ecotone of northern China, which exceeded the effects of local factors[31].

Nevertheless, the impact of patch and landscape factors under fragmentation remains significant, particularly the negative effect of mean of patch area (mean area) at the landscape scale. Likewise, at the patch level, the abundance of soil litter saprotrophic fungi decreased with increased patch size. These results indicate that increased fragmentation can increase the local soil fungal diversity, particularly for foliar endophytes, plant pathogens, animal parasites, and litter saprotrophic fungi. While previous studies often emphasize the negative effects of fragmentation[24,43], it is crucial to recognize that for most biological groups capable of dispersing between patches, the impact can be positive[44]. Fungi typically dispersing through airborne spores are found to be more abundant in isolated fragments compared to well-connected ones[25]. Similar pattern was found by Su et al.[31] across different land-cover types in northern China and by Flores-Rentería et al.[45] in European forest landscapes. Although fragmentation divides continuous landscapes into patches of varying sizes, the presence of matrix and corridors still provides functional connectivity that facilitate fungal dispersal, thereby promoting diversity[24,42]. Additionally, the effects of fragmentation appear independent, with minimal correlations to other local

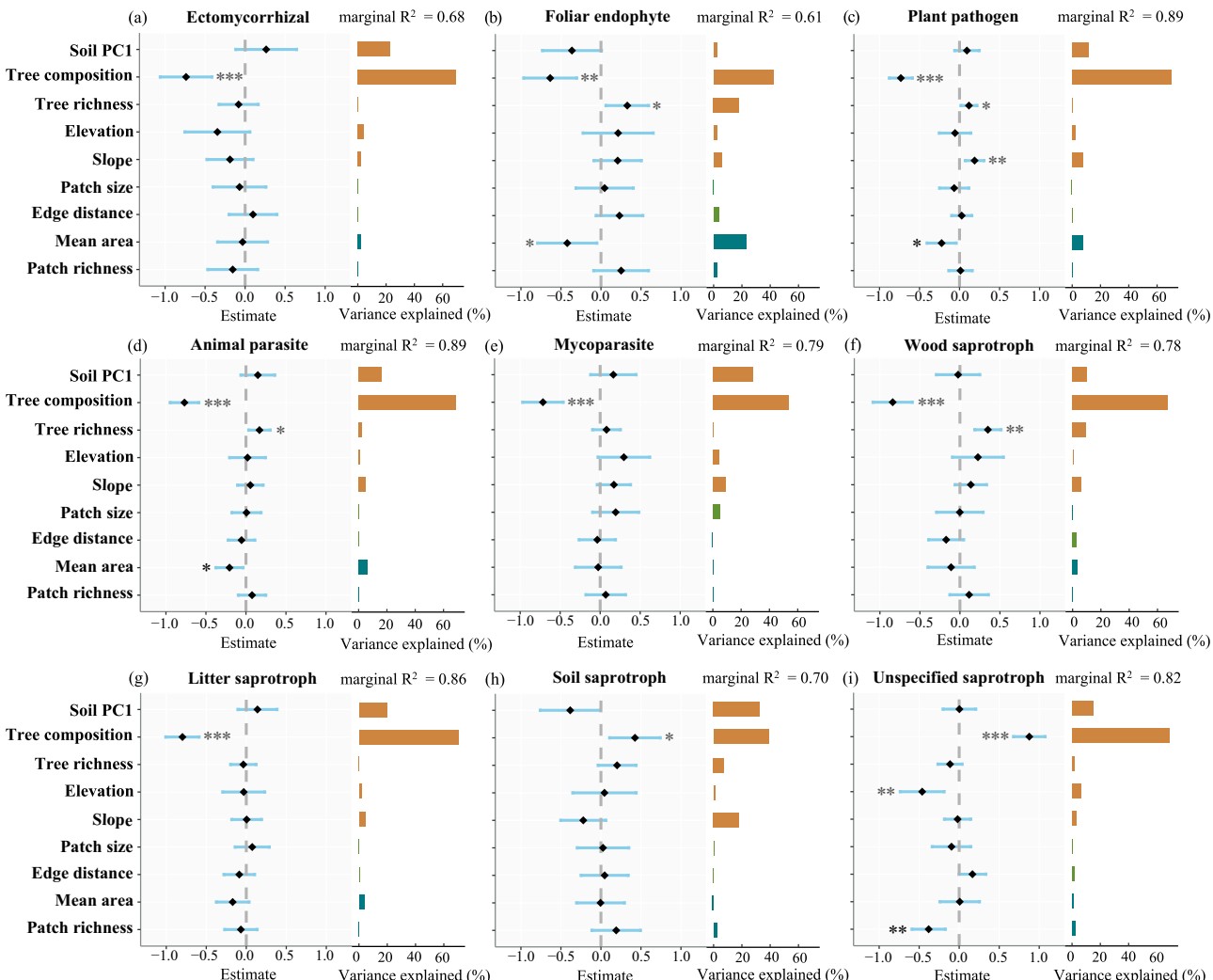

**Fig. 4 | Effects and relative contributions of multi-scale environmental factors on soil fungal community composition.** Panels **a–i** show parameter estimates (corresponding 95% confidence intervals) and explained variance of each predictor for different fungal functional guilds from multivariate linear mixed-effects models (LMMs) ($n = 30$ biologically independent samples). Local factors include soil PC1, tree composition, tree richness, elevation and slope. Patch factors include $\log_{10}$ transformed patch size and edge distance. Landscape factors include $\log_{10}$ transformed mean of patch area (mean area), and patch richness. The significance was tested by $p$-value as *** $p \leq 0.001$; ** $p \leq 0.01$; * $p \leq 0.05$. See Supplementary Data 1 for details and the exact $p$-values.

environmental factors. A similar pattern was demonstrated by a study in the structural equation modeling, with landscape factors influencing soil fungal diversity directly[31].

By integrating multi-dimensional diversity indices, our study provides a comprehensive understanding of how soil fungal diversity responses to environmental factors and demonstrates distinct patterns among functional guilds. We found that the richness and abundance of ectomycorrhizal (ECM) fungi were significantly higher in lowland and montane forests than in limestone forests, which was opposite to other fungal guilds. The main explanation might be that montane forests harbored, on average, 21.5% ECM-associated host plants (Supplementary Table 5), dominated by *Castanopsis fleuryi* and *C. echinocarpa*, along with *Engelhardtia roxburghiana* and *E. serrata*; *Lithocarpus fohaiensis* and *L. fenestratus* were also more abundant than in other forest types[46]. Meanwhile, the dominant species in limestone forests, *Cleistanthus sumatranus* and *Cleidion brevipetiolatum*, are arbuscular mycorrhizal (AM) plants[47], resulting in the lowest ECM fungal diversity. Additionally, the contrary association between richness and abundance of ECM fungi and plant community composition with other fungal functional guilds is likely due to the opposite plant-fungi feedbacks of different fungal guilds. ECM fungi mainly show positive feedbacks with plants in enhancing water and nutrient acquisition for seedlings growth

through their mycelia[2]. In contrast, other functional guilds, like pathogenic fungi and fungal parasites, tend to exhibit negative or neutral feedbacks with plant species[48,49]. Furthermore, the community composition of soil and unspecified saprotrophic fungal communities showed a contrasting pattern across different forest types and plant community composition, compared to other fungal guilds. This may arise from the competition and dynamic balance between fungal guilds. For example, research found that increased abundance of pathogenic fungi could reduce competition ability and diversity of saprotrophic fungi, leading to divergent community responses[50]. In addition, the community composition of foliar endophytes, plant pathogens, and animal parasites responded more strongly to mean patch area than other fungal guilds, likely due to their closer ecological association with host species and greater sensitivity to changes in habitat size and connectivity[23,32]. These results highlight the importance of studying community composition, as shifts in species identities and turnover can have significant effects on host-microbial interactions, plant health, and ecosystem functioning[18].

Finally, one limitation of our study is the underrepresentation of certain fungal functional guilds. In the functional assignment, nearly 70% of OTUs could not be assigned to a specific guild, with nearly one-third unclassified at the phylum level (Supplementary Fig. 1). This limitation may

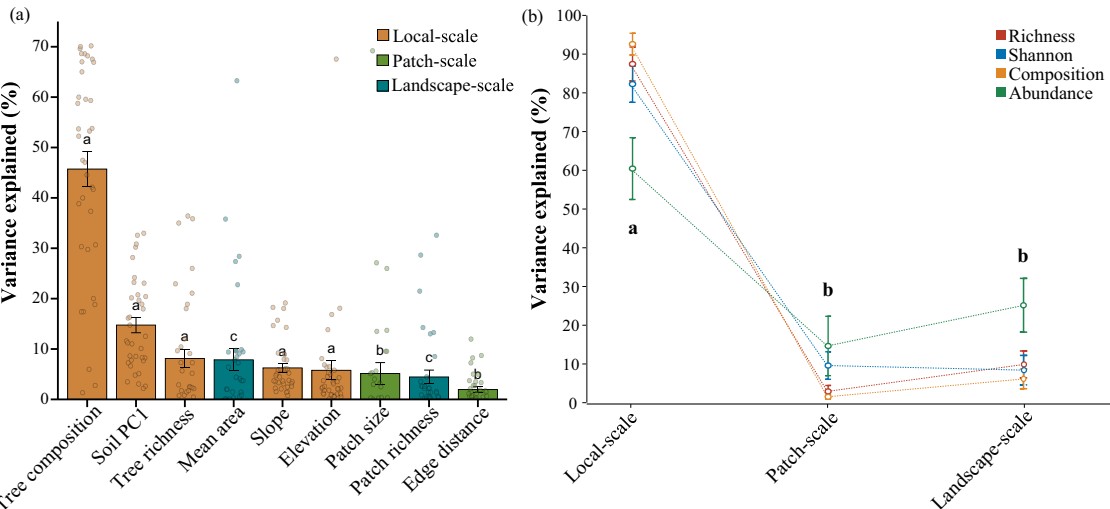

**Fig. 5 | Relative contributions of multi-scale environmental factors to fungal functional guild diversity. a** Mean relative contribution (±s.e.) of each environmental factor (n = 36 biologically independent samples, from nine fungal functional guilds and four diversity indices); and **b** mean relative contribution of local-, patch- and landscape-scale environmental factors to the diversity of different fungal functional guilds (n = 9 fungal functional guilds). Differences among scales were tested using t-tests, and the two guilds with the different letters indicate significant differences (p ≤ 0.05).

partly result from the exceptionally high species diversity and variability of tropical forests, where many taxa remain poorly detected or functional annotated compared with those in temperate regions[8]. Such taxonomic uncertainty restricts a deeper understanding of functional patterns, and future studies could integrate multi-omics approaches or expand reference databases to improve functional resolution[8,51]. In addition, some guilds such as AM fungi, dung and pollen saprotrophs, lichen and protistan parasites, and root endophytes, were detected at very low sequencing abundance and thus excluded from statistical analyses. Other guilds, including mycoparasites and leaf endophytes were retained in the analyses but still exhibited relatively low abundance. These results may partly reflect true ecological rarity, but also suggest potential detection biases of eDNA-based metabarcoding. Future studies should consider targeted sampling, optimized primers, or deeper sequencing to improve the detection of these guilds and better capture the functional diversity of soil fungi in tropical forests[52].

In conclusion, our results highlight the primary role of local tree composition in shaping the diversity of dominant fungal functional guilds in fragmented tropical forests of Yunnan, with patch- and landscape-scale factors having supplementary effects. By capturing strong environmental contrasts across these fragmented tropical systems, our study helps fill a key gap in fungal landscape ecology, an area that remains underrepresented in global biodiversity research. These insights offer several implications for soil fungal diversity prediction and conservation. First, preserving tree species heterogeneity, such as keeping different forest types, is crucial for supporting diverse fungal guilds. Second, smaller patch area does not necessarily reduce soil fungal diversity; instead, maintaining connectivity of patches is essential for supporting fungal dispersal and persistence[53]. Finally, conservation efforts should consider the distinct responses of different fungal functional guilds, as captured through multi-dimensional diversity indices that reflect their ecological roles. This framework is applicable to broader biodiversity prediction across complex ecosystems.

## Methods
### Study region and experimental design
The fragmented forests studied are situated within a 20 km-diameter around the Xishuangbanna Tropical Botanical Garden, Chinese Academy of Sciences, Yunnan Province, southwestern China. Xishuangbanna is part of the Indo-Burma biodiversity hotspot. The annual mean temperature of this region is about 21.5 °C and annual mean precipitation about 1563 mm, with around 80% of the rain falling during the rainy season from May to October[32]. Between 1976 and 2008, the expansion of

fast-growing economic species, particularly rubber plantations (*Hevea brasiliensis*), led to rapid deforestation, reducing natural forest cover from over 70% to 44%[24]. Currently, rubber plantations dominate more than 64% of the studied landscape, leaving only 25% as remaining natural forests[54].

In 2007, 30 survey plots spanning a gradient of fragmentation were established to evaluate the consequences of fragmentation on biodiversity in Xishuangbanna Dai Autonomous Prefecture, Yunnan Province, southwestern China[54,55]. These plots, spanning 17 patch sizes, were classified into three forest types: limestone, lowland, and montane, based on ecological, morphological, and composition of vegetation characteristics (Supplementary Fig. 9). For larger patches, we established a greater number of sampling plots to adequately capture the environmental variations associated with edge distance and forest type. Each plot consisted of circular subplots with a 5-meter diameter, spaced ten meters apart along the length of the fragment sites. The number of subplots per sampling plot varied from 5 to 15, as we used a variable-area method[55]. This approach, with the number of subplots adjusted according to tree density, ensured that at least 100 trees (diameter at breast height ≥ 1 cm) were recorded in each plot (Supplementary Table 6)[32].

### Soil sampling and soil properties analysis
In May 2021, we collected soil samples from all the 30 plots, using a five-point sampling method. Specifically, five soil cores (5 cm in diameter and 10 cm in depth) were collected at equal distances radiating outward from the center of each plot, following the arrangement directions of the subplots (Supplementary Fig. 10). The five cores from each plot were combined into a single sample (approximately 1200 g of soil) and transported to the laboratory on the same day. In the lab, roots and stones were carefully removed using a 2 mm soil sieve which was cleaned, sterilized, and dried before processing. Fifty grams of root-free soil from each sample were reserved for DNA sequencing and stored at −20 °C, with the remaining kept at 5 °C for analyzing soil physicochemical properties.

The soil properties were measured for each plot. Soil pH was measured with a pH meter using a 1:2.5 soil-to-water ratio. Soil organic matter (OM) was determined by dichromate oxidation method, which involves oxidizing soil organic carbon with potassium dichromate and sulfuric acid, followed by titration of the remaining dichromate to estimate soil OM content[56]. Soil total N, P, potassium (K), and calcium (Ca) were measured using an Elementar Vario EL III elemental analyzer (Hannover, Germany).

## Molecular and bioinformatic analysis

Fungal DNA was extracted from 0.5 g of soil using the DNeasy PowerSoil Kit (Mo Bio Laboratories, Carlsbad, CA, USA). The internal transcribed spacer 1 (ITS1) region was amplified using the primer set ITS1F and ITS1R[57], with a 12 bp barcode added to the reverse primer for each sample. To minimize polymerase chain reaction (PCR) bias, the ITS1 region was amplified in a 50 μl reaction containing 25 μl of 2× Premix Taq (Takara Biotechnology, Dalian Co. Ltd., China), 1 μl of each primer, and 3 μl of DNA template (20 ng/μl). The PCR conditions were as follows: initial denaturation at 94 °C for 2 min, followed by 25 cycles of 94 °C for 30 s, 55 °C for 30 s, and 72 °C for 1 min, with a final extension at 72 °C for 10 min. DNA extraction and PCR amplification were each performed once per soil sample for fungal analysis. Negative controls were included for both steps to monitor potential contamination, and no detectable amplification was observed by gel electrophoresis. PCR products from each sample were pooled, purified, and used to construct sequencing libraries, which were sequenced using the Illumina MiSeq platform (2 × 250 bp paired-end reads; Illumina, San Diego, CA, USA).

In total, 2,941,702 raw sequences were generated from all samples, and were retained for downstream analysis. The paired-end reads from all the 30 soil samples were quality-filtered and assembled using the FASTX-Toolkit (Hannon Lab). Potential chimeras and singletons were removed using UPARSE, and sequences with a similarity threshold ≥97% were clustered into operational taxonomic units (OTUs)[58]. Representative sequences of all fungal OTUs were classified using the Ribosomal Database Project (RDP) classifier[59], and matched against the UNITE database[52]. In total, 1,198,198 sequences were identified and clustered into 8288 OTUs. We rarefied each sample to the smallest sequence count observed (23,991 sequences) to eliminate the impact of sequencing depth and enable comparison among samples (Supplementary Fig. 11). Using the expert-curated FungalTraits database[7], we categorized fungal species into functionally distinct guilds: animal parasites, ectomycorrhizal fungi, foliar endophytes, litter saprotrophs, mycoparasites, plant pathogens, soil saprotrophs, unspecified saprotrophs, and wood saprotrophs (guilds with fewer than 10 sequences in the plot were excluded due to insufficient representation). Representative sequences have been submitted at NCBI (SRA PRJNA1171718).

## Environmental factors collection

To investigate the impact of local-scale factors on soil fungal diversity, we considered soil properties, tree species diversity of richness, Shannon diversity and community composition, elevation and slope. We measured the richness and abundance of tree species (diameter at breast height ≥ 1 cm) in each plot (i.e., the total tree species and individuals in subplots; Supplementary Fig. 10). Tree community composition (hereafter 'tree composition') was represented by the first axis of Principal Coordinates Analysis (PCoA) from abundance-based Bray-Curtis dissimilarity using the *vegan* package (V2.6–6.1)[60]. In addition, we determined the mycorrhizal type of all recorded plant species using the FungalRoot v2.0 database[46], and recorded the presence of ectomycorrhizal (ECM) host species in each plot (Supplementary Table 5). Topographic variables, including elevation and slope, were measured on site. Finally, the soil properties included soil pH, OM, N, P, K, and Ca content.

To assess the impact of patch and landscape factors on soil fungal diversity, we considered patch-scale factors of patch size and edge distance, and landscape-scale factors of mean of patch area (mean area), edge density, patch density, and patch richness within a 2 km spatial radius, using images from the China's Land-Use/Cover Datasets (CLCD, 30 m resolution)[61]. Patch size is the exact area of the patch corresponding to the sampling plot, and edge distance is the Euclidean distance from the plot to the nearest patch edge. The mean of patch area (mean area) measures the landscape composition by averaging the area of all patches within the landscape. Edge density describes the landscape configuration by calculating the total length of all edge segments relative to the overall landscape area. A landscape with a higher aggregation of classes will exhibit a lower edge density. Patch density

is the ratio of the number of patches to the overall landscape area. We chose a 2 km radius to calculate landscape factors, as it ensures all sampling points have neighboring patches and has been identified as a key distance affecting belowground microorganisms[30]. The landscape factors were calculated using the *landscapemetrics* package (V2.1.2)[62].

## Data analysis

We calculated four diversity indices for each fungal functional guild using the *vegan* package (V2.6–6.1)[60]. These indices include OTU richness (i.e., the number of fungal OTUs identified in each soil sample), Shannon diversity (considers both OTU richness and evenness), community composition (represented by the first axis of the PCoA based on Bray-Curtis dissimilarity), and abundance (the number of sequences for each fungal functional guild after rarefaction).

For the environmental factors measured at different scales, we first compared the Spearman rank correlations among soil properties using the *corrplot* package (V0.92)[63]. Due to the high correlations between soil properties (Supplementary Table 7), we conducted a principal component analysis (PCA) on all measured soil variables, extracting the first axis to represent the overall variation in soil conditions (i.e., soil PC1). Higher PC1 values indicate higher soil pH, N, OM, P, and Ca content, with lower K content. Additionally, we log$_{10}$ transformed patch- and landscape-scale factors of patch size, edge distance, mean area, edge density, patch density, and patch richness to ensure the normal distribution of predictors. We also conducted a Spearman rank correlation among all the local-, patch- and landscape-scale factors, and excluded the highly correlated variables with correlation ≥0.7 in further analysis. Since forest type can shape the regional environmental factors, we treated forest type as the independent variable and compared the variation in local (soil PC1, tree composition, tree richness, elevation, slope), patch (patch size, edge distance), and landscape (mean area, patch richness) factors among forest types using a one-way analysis of variance (ANOVA).

To explore the relationships between environmental factors and diversity of each fungal guild, we first performed one-way ANOVA, treating forest type as the independent variable, to compare the diversity and composition (dependent variables) of each fungal functional guild across different forest types. We then used the Spearman rank correlation analysis to examine the strength of correlations (*r*-value) and significance (*p*-value) between all the environmental factors (except for forest types) and the diversity/composition of each fungal functional guild. Given that different forest types and fragment patch can influence the effects of environmental factors in the region, we built multivariate linear mixed-effects models using the *lmerTest* package (V3.1–3)[64]. In these models, different patches and forest types were included as random effects, with all the environmental factors as predictor variables (Spearman's correlation ≤ 0.7 and a variance inflation factor [VIF] ≤ 4), and four diversity indices of each fungal functional guild as the response variables, respectably. All predictor and response variables were standardized (*z*-scored) to allow comparison. We assessed model fit using the AIC$_c$ (small-sample corrected Akaike Information Criterion), and calculated the relative contribution of each predictor using the *glmm.hp* package (V0.1–3)[65]. Model selection based on AIC$_c$ was conducted to identify the optimal combination of variables affecting the diversity indices of each fungal guild, keeping the model with the smallest AIC$_c$ value. Spatial autocorrelation was assessed by Moran's *I* function (based on the 9 nearest neighbors) using the *spdep* package (V1.3–10)[66], with no significant results in all models (Supplementary Table 8). The correlation between the diversity/composition of each fungal functional guild and individual soil properties was evaluated using Spearman rank correlation analysis. Additionally, the relationship between the fungal diversity/composition and tree composition was further explored, with the four diversity indices of each fungal functional guild fitted to the two axes of PCoA of tree community composition.

## Statistics and reproducibility

All data analyses were performed using R software (version 4.2.0)[67]. Each of the 30 forest plots represented an independent biological replicate, with five composited soil cores forming one representative sample per plot. Fungal functional guild was categorized using the FungalTraits database. Fungal diversity indices (richness, Shannon diversity, community composition, and abundance) were calculated from rarefied OTU tables. Differences among forest types were tested by one-way ANOVA with TukeyHSD test. Spearman correlations assessed associations between environmental variables and fungal diversity. Linear mixed-effects models evaluated the effects and relative contributions of predictors, including forest type and patch identity as random effects. Multicollinearity (VIF ≤ 4) and spatial autocorrelation (Moran's *I*) were checked to ensure model robustness.

## Reporting summary

Further information on research design is available in the Nature Portfolio Reporting Summary linked to this article.

## Data availability

Raw sequencing data have been deposited in the NCBI SRA under accession code PRJNA1171718. All other data supporting the findings of this study, including the source data for Figs. 1–5 and the taxonomic and functional guild classification files, are available on Figshare at https://doi.org/10.6084/m9.figshare.28642496. Detailed results underlying Fig. 4 and Supplementary Fig. 6–8 are provided in Supplementary Data 1.

## Code availability

The code used in this study is available on Figshare at https://doi.org/10.6084/m9.figshare.28642496 in the file code.R.

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

## Acknowledgements

The work was supported by the National Natural Science Foundation of China (32301426), the China Postdoctoral Science Foundation (2024T170025). We thank the Xishuangbanna Tropical Botanical Garden, CAS for providing the experimental platform and Xishuangbanna National Nature Reserve for assistant in sampling collection guidance. We greatly thank Jiajia Liu for the guidance on sampling design and contribution in the field work. We thank Kyle W. Tomlinson, Yun Lu, Zongze Yang, Xiaomao Wang, Yiran Zhao, Xianhui Shen and Minglong Liu and all the lab members for great assistance in the field and lab work.

## Author contributions

Y.L. and S.Z. conceived and designed the study. Y.L., S.Z. and Y.C. collected the data. Y.L., J.H. and Y.C. performed the analyses. Y.L. and Y.C. wrote the first draft of the paper and all authors revised the paper carefully.

## Competing interests

The authors declare no competing interests.
