## [Transparent Peer Review file · Communications Biology]

Cross-scale drivers of soil fungal diversity in fragmented forests of southwestern China

Corresponding Author: Dr Ying Chen

Version 0:

Reviewer comments:

Reviewer #1

(Remarks to the Author)

The manuscript titled "Cross-scale drivers of soil fungal diversity in fragmented forests" by Yawen Lu and colleagues investigates how environmental factors operating at different spatial scales—local, patch, and landscape—shape the diversity and composition of soil fungi in fragmented tropical forests in Xishuangbanna, southwestern China.

To address this, the researchers analyzed soil samples from 30 plots distributed across 17 forest fragments representing three distinct forest types: limestone, lowland, and montane. Using high-throughput sequencing of the ITS1 region, they identified and categorized soil fungal communities into functional guilds. Environmental data were collected at three spatial scales: local factors such as tree diversity and soil properties; patch-level variables including patch size and distance from edges; and landscape-level metrics like patch richness and mean patch area within a 2 km radius.

The study found that local tree composition was the most significant driver of soil fungal diversity and community composition, explaining nearly 46% of the variation observed. This influence remained dominant across all fungal functional groups and diversity indices, such as species richness, Shannon diversity, community composition, and abundance. While soil properties, elevation, and tree richness also played roles, their effects were comparatively modest.

The study concludes that local environmental factors, especially tree community composition, are the primary determinants of soil fungal diversity in fragmented forests.

1.) I want to start by highlighting that this is a very interesting and novel study that I believe will be of great interest to the audience of the journal. The study covers a very interesting concept and it is clear that the authors have poured a great deal of work both in the study design and in collecting the relevant data. The study design is sound and appears reproducible and works well to answer the research questions that the authors lay out. Lastly I want to commend the authors for their in depth statistical analysis and for the visually impressive and informative graphs, both in the main text and in the supplementary information.

2.) However, I do have some concerns that currently hold the manuscript back from being fully ready for publication. My biggest concern is that the grammar, syntax and overall use of the English language is not up to publication standards and needs a thorough revision. There are even some cases where I am not sure about the meaning/message that the sentences are trying to convey. As I believe this manuscript to be a very interesting and valuable piece of work, I have the utmost confidence that the authors can correct the mistakes and elevate the language of the paper to a level that is appropriate for publication. Below I will point out several places where this is needed.

-Line 37: Better to say "the results show"

-Line 39-41: I guess you wanted to say "even though"? If not, revise this sentence a bit as it reads awkwardly at the moment. Also delete the space before the period.

-Line 42: Switch patten to patterns

-Line 42-45: This sentence reads awkwardly. Maybe it is best to split it to 2 sentences? I m not sure what it is trying to say.

-Line 48: better to say conservation efforts

-Line 53-54: change to ARE among the largest contributors to global biomass AND play...

-Line 62-63: change to "As fungi are classified into multiple functional types and differ in their nutritional habits..."

-Line 71: Delete "the"

-Line 82-83: This reads awkwardly. The factors themselves are not the habitat but rather the soil. This should be reworded.

-Line 84: Change Whereas to However

- Line 104" Change to "As some fungal guilds have the ability to disperse across larger scales" after "Moreover"
- Line 109-110: Change to "Of a habitat's spatial configuration after..." after "As the alteration".
- Line 110-111: "it is likely for soil fungi reducing potential colonization and persistence in the remaining habitat patches". I am not sure what this sentence means. Is it saying that soil fungi will likely reduce their potential colonization rates and persistence in the habitat patches? If so, it needs rewording as right now it is not correct in terms of syntax and grammar.
- Line 167: Change "for" to "from"
- Line 172: change "was" to "were". Keep in mind for the rest of the manuscript that grams is plural and not singular even if you are referring to the grams of a single soil sample.
- Line 233-234: change to "using the vegan package..." and mention also the version.
- Line 257: change "employed the" to "used"
- Line 269: I assume you wanted to say "the end"?
- Line 271-272: change "with the model having the smallest AICc value kept" to "Keeping the model with the smallest AICc value".
- Line 295-298: Split this into two sentences. It reads a bit awkward
- Line 316-319: I don't understand this the way that this is written. What is positively correlated in the end?
- Line 325: change "increased" to "increasing" and "decreased" to "decreasing"
- Line 326: remove "mainly"
- Line 331: change "was" to "were"
- Line 343: change to "negatively associated with"
- Line 353-354: Something cannot reach the highest of explaining. This needs to be rephrased
- Line 451: change to "A similar pattern"
- Line 452: "demonstrated by A study"
- Line 458: "Pick either "of" or "in"
- Line 473: Switch "group" to plural.

3.) Line 140-144: The research questions you lay out are sound and clear and logically follow the introduction. However, the paper is currently lacking any hypotheses. I understand that this topic is understudied but I think based on what you said in your introduction and some of the mentioned literature, you can formulate some hypotheses for your questions.

4.) Your study design is very interesting and there is a commendable amount of work put into the experiments and data collection. However, the materials and methods is currently lacking clarity and needs more work. One general concern that I have with the materials and methods is that it is not at all clear to me how many replicates you have of what. In general this section needs some work to provide more details of the experimental methods with a more logical flow of all the processes. It is very important for the reader to know your replicates that then go into your statistics. Below I highlight some instances where the materials and methods is unclear and some suggestions. Please apply these also to the rest of the section in case I have missed something:

- Line 148-158: I would even consider moving this entire paragraph to supplementary information. It makes the materials and methods larger without adding much to it and people can always download your SI if they want to read it.
- Line 159-165: I would start with this paragraph in your materials and methods.
- Line 164: How many circular subplots?
- Line 167-168: This here is why its important to mention before how many subplots were in each plot. Because now I am wondering, how many did you collect? Was it 5 per subplot? If there were 5 subplots was it then 25 per forest? Etc etc. Be detailed about it.
- Line 169-171: How many grams total? 50grams seems a lot for DNA extraction if you re using a soil kit that requires such little grams to work with. How many times was each replicated for the DNA?
- Line 175-202: This reads fine but it's a little bit abrupt following from the previous paragraph. The last thing you mention is the physiochemical properties which I expected to see right below but instead it then jumps over to DNA extractions. I propose that you move the text with physiochemical properties here and then have the DNA extractions and sequencing details on a separate section right after.
- Line 211: The soil properties were measured, not collected.
- Line 212-213: I Am not familiar with this. Can you describe this method? Typically studies use Loss on ignition. If this is another common method better to briefly describe it or provide a ref that does it.
- Line 232: Again without knowing your number of replicates from the M&M it becomes hard to assess the validity of the statistics. They seem well thought out but replicates are important to mention.
- Line 248-254: Was forest the independent variable here? Make it clear
- Line 254: What kind of ANOVA? One way? two-way?
- Line 255-257: What is the dependent and independent variable?

5.) The documentation of some results needs improvement (e.g. anova test results sometimes referred to as correlations). Below I highlight some of these:

- Line 289-290: Be careful how you phrase your results. Here you are referring to your graph in supplementary information which is a boxplot showing your anova results. This is no longer spearman correlations and should not be referred to as such.
- Line 292: Again this is an anova right? This sounds like a correlation. Maybe its better to say that forest type significantly affected/shaped fungal diversity?
- Line 299-302: I might be mistaken but when you talk about composition and dissimilarity isnt it then better to show this via a PCoA? If you choose a boxplot maybe its better to describe it differently.

Other comments:

- 6.) Line 57: What are soil carbon resources? Do you mean soil carbon in general? Then maybe its better to not call it carbon resources but just soil carbon
- 7.) Line 58: Maybe its worth to switch Mycorrhizal to "Beneficial symbiotic fungi" to keep it more broad at an introduction level.
- 8.) Line 70: Why biotic or abiotic? Dont both interact? I would then say biotic AND abiotic
- 9.) Line 75-76: Is that always the case? I think it is also good to address older literature that says the opposite (acidic pH favoring fungi) as well like:
 - Fungal biomass development in a chronosequence of land abandonment - van der Wal et al., 2005
 - Site and management effects on soil microbial properties of subalpine meadows: a study of land abandonment along a north-south gradient in the European Alps - Zillet et al., 2001
- 10.) Line 77: I would abbreviate nitrogen to N after the first time you mention it and across the rest of the document. Similarly for other nutrients/metals etc
- 11.) Line 87-88: Is there a reference to support this?
- 12.) Line 88-89: Give maybe a small definition of edge effect in this context in a parenthesis to make this more accessible
- 13.) Line 117: Which mixed effects? If you are mentioned THE mixed effects I would expect that the previous sentence mentions some of them but the previous sentence just mentions strong effects with no indication of them being mixed.
- 14.) Line 213: For these two (N and P) as I mentioned you can already abbreviate them earlier in the text when you first mention them
- 15.) Line 364-377: This needs to be condensed a bit. It's a full paragraph of just summarizing results that we just read. Its good to start the discussion with a brief results summary but this is a bit too long.
- 16.) Line 406-407: If I m not mistaken, in the intro you said that about basic soils and I pointed out about acidic so you have to be consistent.
- 17.) Line 440-441: Cite some of those previous studies here

Reviewer #2

(Remarks to the Author)

Dear Authors,

I did enjoy Reading your manuscript entitled "Cross-scale drivers of soil fungal diversity in fragmented forests", moreover exploring this question in the region of Yunnan. The manuscript is clear, rich, well illustrated and explores the importance of distinct factors across scales.

I have a few remarks – and it will be a nice example for future research on soil fungi and their ecology. It fits the scope of Communications Biology, represents an inspiring and importance peace of work. I would recommend to better contextualize the conclusions notably as you mentioned the tropical context - where landscape ecology of fungi is poorly studied. On this aspect, compared to most studies you mention from fragmented forests, your habitats are much more contrasted, and it could be mentioned.

Indeed, the differences in forest type - that could be considered as distinct habitats from a mycological point of view - shall be better introduced, as a contrasted factor to compare the importance of landscape factors.

You may also discuss some limits - especially when discussing the response of some low represented guilds (while all fungi response was not al all included). Perhaps you could also strengthen the importance of considering the impact of fragmentation not only on species richness but also other indicators, as it's also one original aspect of this study, and it extends beyond fungi ecology ?

More punctual comments :

First, the forest habitat you choose is particularly contrasted, with these three forest types that could be considered as distinct habitats based on soils and tree composition, especially from a mycological point of view. Your figures 1 and S3 illustrate well these contrasts, and so do the results. It would be important to clarify it for the readers in the introduction, and in the analysis (whether forest type was included in the models or not). Please see line per line suggestion where to detail this.

In the MM, you mention the description of tree composition, but in the discussion name some AM species – not ectomycorrhizal (ECM) hosts. However, given the high abundance of ECM in your dataset, why not naming these species of including a factor relative to ECM host species number? You have already done many analyses – I do not ask to make more, but at least mention it in the MM and the discussion.

In the MM - for molecular biology part, negative controls of extraction, PCR, sequencing are not mentioned; nor the number of PCR as you mentioned pooling some products. I would recommend describing in the results a bit more this dataset, it's classical with metabarcoding data and would also reflect what are these fungal communities in tropical fragmented forests. For example, did you have a high heterogeneity of abundances before rarefying ? How many sequences did you lose during the bioinformatic analyses ? Based on rarefaction, you keep only half of your dataset, which is not trivial. It would be important to mention it at least.

Based on figure S4, we can observe that all sequences were not functionally assigned which is normal in any soils. As you did not test a model on all soil fungi – it's important to avoid conclusions on all fungi, especially as these unknown fungi represent a large part of the sequences in limestone sites. Moreover, at least in supplementary, it would be important to evaluate the representativity of each guild, in term of abundance and OTU number (a table ?). Mycoparasite, endophytes look low abundant, and it can be discussed as a limit of metabarcoding on soil eDNA. Considering that you discuss the patterns among different guild, I would suggest keeping Figure S4 in the main document. You could have in supplementary a figure on the taxonomic diversity – to also discuss how far the level of assignation was limitant or not to distinguish the functional guilds.

Still in the methods, I was wondering how distance was taken into account, if spatial autocorrelation was significant or not ? It 's a basic test you could add – to be sure it's not significant, and that your closely related plots are independent. If you do not want to include it as you removed correlation later, please justify this approach.

Finally, the impact of fragmentation is often evaluated on alpha diversity changes – loss of species, increase or decrease in diversity. Interestingly, you also evaluated changes in composition – and found several relationships with patch size notably. This is not really introduced while it's an important result of your study, especially as you chose to put Figure 3 in principal and not the S7.

Additional comments :

See the main documents for line per line suggestions.

Table S5 : Is there a significance for the colors in the table ?

Best
Mélanie Roy

Version 1:

Reviewer comments:

Reviewer #1

(Remarks to the Author)

The manuscript titled "Cross-scale drivers of soil fungal diversity in fragmented forests" by Yawen Lu and colleagues investigates how environmental factors operating at different spatial scales—local, patch, and landscape—shape the diversity and composition of soil fungi in fragmented tropical forests in Xishuangbanna, southwestern China.

To address this, the researchers analyzed soil samples from 30 plots distributed across 17 forest fragments representing three distinct forest types: limestone, lowland, and montane. Using high-throughput sequencing of the ITS1 region, they identified and categorized soil fungal communities into functional guilds. Environmental data were collected at three spatial scales: local factors such as tree diversity and soil properties; patch-level variables including patch size and distance from edges; and landscape-level metrics like patch richness and mean patch area within a 2 km radius.

The study found that local tree composition was the most significant driver of soil fungal diversity and community composition, explaining nearly 46% of the variation observed. This influence remained dominant across all fungal functional groups and diversity indices, such as species richness, Shannon diversity, community composition, and abundance. While soil properties, elevation, and tree richness also played roles, their effects were comparatively modest.

The study concludes that local environmental factors, especially tree community composition, are the primary determinants of soil fungal diversity in fragmented forests.

I really enjoyed reading your revised manuscript after my vacation. I am very happy to see that the authors have meticulously revised the manuscript and addressed all points of concern from both reviewers. I believe that the manuscript now clearly conveys the intended message and is up to the publication quality standards of the journal.

Although the comments were many and quite tough in some places, the authors have done a fantastic job of incorporating the feedback.

I am very satisfied with the review and have no further concerns that need attention from my side. Good luck in your future endeavors.

Reviewers' comments:

Reviewer #1 (Remarks to the Author):

The manuscript titled "Cross-scale drivers of soil fungal diversity in fragmented forests" by Yawen Lu and colleagues investigates how environmental factors operating at different spatial scales—local, patch, and landscape—shape the diversity and composition of soil fungi in fragmented tropical forests in Xishuangbanna, southwestern China.

To address this, the researchers analyzed soil samples from 30 plots distributed across 17 forest fragments representing three distinct forest types: limestone, lowland, and montane. Using high-throughput sequencing of the ITS1 region, they identified and categorized soil fungal communities into functional guilds. Environmental data were collected at three spatial scales: local factors such as tree diversity and soil properties; patch-level variables including patch size and distance from edges; and landscape-level metrics like patch richness and mean patch area within a 2 km radius.

The study found that local tree composition was the most significant driver of soil fungal diversity and community composition, explaining nearly 46% of the variation observed. This influence remained dominant across all fungal functional groups and diversity indices, such as species richness, Shannon diversity, community composition, and abundance. While soil properties, elevation, and tree richness also played roles, their effects were comparatively modest.

The study concludes that local environmental factors, especially tree community composition, are the primary determinants of soil fungal diversity in fragmented forests.

1.) I want to start by highlighting that this is a very interesting and novel study that I believe will be of great interest to the audience of the journal. The study covers a very interesting concept and it is clear that the authors have poured a great deal of work both in the study design and in collecting the relevant data. The study design is sound and appears reproducible and works well to answer the research questions that the authors lay out. Lastly I want to commend the authors for their in depth statistical analysis and for the visually impressive and informative graphs, both in the main text and in the supplementary information.

[Response] We sincerely thank you for your kind and encouraging feedback. We truly appreciate your recognition of the importance and novelty of our research, as well as the

work invested in the study design, data collection, statistical analysis and presentation. These comments are highly valued and truly motivating for our team.

2.) However, I do have some concerns that currently hold the manuscript back from being fully ready for publication. My biggest concern is that the grammar, syntax and overall use of the English language is not up to publication standards and needs a thorough revision. There are even some cases where I am not sure about the meaning/message that the sentences are trying to convey. As I believe this manuscript to be a very interesting and valuable piece of work, I have the utmost confidence that the authors can correct the mistakes and elevate the language of the paper to a level that is appropriate for publication. Below I will point out several places where this is needed.

[Response] We sincerely appreciate this helpful comment. We fully agree that clear and proper language is crucial for communicating our findings. We have now carefully revised the entire manuscript to improve grammar, syntax, and readability and believe these improvements have considerably elevated the language quality of the manuscript.

-Line 37: Better to say "the results show"

[Response] Thank you. Changed.

-Line 39-41: I guess you wanted to say "even though"? If not, revise this sentence a bit as it reads awkwardly at the moment. Also delete the space before the period.

[Response] Thank you very much for pointing this out. We have now revised the sentence and deleted the space. It now reads:

"Additionally, although the relative contributions of patch and landscape factors were lower than those of local factors, the mean patch area at the landscape scale still showed a significant negative association with the diversity of fungal functional guilds" in lines 39–42.

-Line 42: Switch patter to patterns

[Response] Thank you. Done (line 43).

-Line 42-45: This sentence reads awkwardly. Maybe it is best to split it to 2 sentences? I m not sure what it is trying to say.

[Response] Thank you very much. We have revised the sentence and it now reads:

"Some fungal functional guilds showed contrasting patterns to environmental factors. Specifically, the richness and abundance of ectomycorrhizal fungi exhibited opposite trends to forest types relative to other guilds, whereas the community composition of soil/unspecified saprotrophic fungi showed contrasting responses to tree composition compared with other guilds." in lines 42–47.

-Line 48: better to say conservation efforts

[Response] Thank you. Revised (line 50).

-Line 53-54: change to ARE among the largest contributors to global biomass AND play...

[Response] Thank you. Changed (line 55).

-Line 62-63: change to "As fungi are classified into multiple functional types and differ in their nutritional habits..."

[Response] Thank you. Changed (lines 65–66).

-Line 71: Delete "the"

[Response] Thank you. Done (line 76).

-Line 82-83: This reads awkwardly. The factors themselves are not the habitat but rather the soil. This should be reworded.

[Response] Thank you for the suggestion. We have now changed the sentence to:

"Given their direct influence on soil fungi and mediation of belowground trophic processes, local-scale environmental factors are widely considered in microbial diversity predictions (Tedersoo et al., 2014)." in lines 89 – 91.

-Line 84: Change Whereas to However

[Response] Thank you. Changed (line 91).

-Line 104" Change to "As some fungal guilds have the ability to disperse across larger scales" after "Moreover"

[Response] Thank you. Changed (line 118).

-Line 109-110: Change to "Of a habitat's spatial configuration after..." after "As the

alteration".

[Response] Thank you. Changed (lines 123–124).

-Line 110-111: "it is likely for soil fungi reducing potential colonization and persistence in the remaining habitat patches". I am not sure what this sentence means. Is it saying that soil fungi will likely reduce their potential colonization rates and persistence in the habitat patches? If so, it needs rewording as right now it is not correct in terms of syntax and grammar.

[Response] Thank you. Your understanding is correct and we have revised the sentence. It now reads:

"...it may reduce the potential for soil fungi to colonize and persist in the remaining habitat patches." in lines 124 - 125.

-Line 167: Change "for" to "from"

[Response] Thank you. Changed (line 193).

-Line 172: change "was" to "were". Keep in mind for the rest of the manuscript that grams is plural and not singular even if you are referring to the grams of a single soil sample.

[Response] Thank you for your suggestion and we have revised carefully (line 201).

-Line 233-234: change to "using the vegan package..." and mention also the version.

[Response] Thank you for the suggestion. We have revised the expression and subsequent related content.

-Line 257: change "employed the" to "used"

[Response] Thank you. Changed (line 299).

-Line 269: I assume you wanted to say "the end"?

[Response] Sorry, "and" is redundant, we have deleted it (line 311).

-Line 271-272: change "with the model having the smallest AICc value kept" to "Keeping the model with the smallest AICc value".

[Response] Thank you. Changed (line 314).

-Line 295-298: Split this into two sentences. It reads a bit awkward

[Response] Thank you for your suggestion. It has now been revised to:

" *For Shannon diversity, the patterns of different fungal functional guilds were generally similar. Limestone forests showed significantly higher Shannon diversity than lowland forests (for ectomycorrhizal fungi and mycoparasites) and montane forests (for all fungal functional guilds) (Fig. 2b).*" in lines 254 - 358.

-Line 316-319: I don't understand this the way that this is written. What is positively correlated in the end?

[Response] Thank you for this insightful comment and sorry for the vague. In our analysis, "positively correlated" means that the variation in fungal community composition (PCoA1 scores) changes in the same direction as the variation in tree community composition (PCoA1 scores) along the primary ordination axis. Importantly, the positive/negative sign here refers to the relative direction of change in multivariate space (more intuitive in Supporting Information Figure S7), and does not imply an ecologically "positive" or "negative" effect. To address this concern, we revised the corresponding result description (lines 378–388) to read as follows:

"Multivariate linear mixed effect models (patches and forest types as random effects) verified significant effects of tree composition (PCoA1) on the diversity of nine fungal functional guilds. The community composition (PCoA1) of all fungal functional guilds was significantly affected by tree composition. Specifically, saprotrophs (standardized regression coefficient [β] = 0.425, p = 0.017) and unspecified saprotrophs (β = 0.876, p < 0.001) showed positive relationships with tree community composition, indicating that their compositional variation followed the same gradient as tree communities (Fig. 4 and Table S5). In contrast, other fungal functional guilds displayed negative relationships, meaning their compositional variation occurred in the opposite direction along the tree compositional gradient (Fig. 4 and Table S5)."

-Line 325: change "increased" to "increasing" and "decreased" to "decreasing"

[Response] Thank you. Changed (lines 392–393).

-Line 326: remove "mainly"

[Response] Thank you. Done (line 394).

-Line 331: change "was" to "were"

[Response] Thank you. Changed (line 399).

-Line 343: change to "negatively associated with"

[Response] Thank you. Changed (line 411).

-Line 353-354: Something cannot reach the highest of explaining. This needs to be rephrased

[Response] Thank you for the comment. We have now revised the sentence to:

"In the multivariate linear mixed effect models, tree composition was the strongest predictor, explaining $45.72 \pm 3.49\%$ (mean \pm s.e.) of the variance across the four diversity indices of the nine functional guilds, followed by soil PC1 ($14.75 \pm 1.52\%$) and tree richness ($8.11 \pm 1.88\%$)." in lines 421–424.

-Line 451: change to "A similar pattern"

[Response] Thank you. Changed (line 520).

-Line 452: "demonstrated by A study"

[Response] Thank you. Revised (line 521).

-Line 458: "Pick either "of" or "in"

[Response] Thank you. We kept "in" (line 532).

-Line 473: Switch "group" to plural.

[Response] Thank you. Changed to "guilds" (line 545).

3.) Line 140-144: The research questions you lay out are sound and clear and logically follow the introduction. However, the paper is currently lacking any hypotheses. I understand that this topic is understudied but I think based on what you said in your introduction and some of the mentioned literature, you can formulate some hypotheses for your questions.

[Response] Thank you very much for this helpful suggestion. We fully agree that explicitly stating our hypotheses would further clarify the framework of our study. Based on our introduction and the current literature, we have now added clear hypotheses to the revised manuscript in lines 166–173. It now reads:

"Specifically, we propose the following hypotheses: (i) Local-scale vegetation characteristics and soil properties will be the primary drivers of soil fungal diversity in fragmented forests. (ii) Patch- and landscape-scale factors will also exert significant effects on soil fungal diversity, providing complementary contributions to the variation explained beyond those of local factors. (iii) Different soil fungal functional guilds will exhibit distinct responses to environmental factors across spatial scales, reflecting their distinct ecological strategies and environmental dependencies."

4.) Your study design is very interesting and there is a commendable amount of work put into the experiments and data collection. However, the materials and methods is currently lacking clarity and needs more work. One general concern that I have with the materials and methods is that it is not at all clear to me how many replicates you have of what. In general this section needs some work to provide more details of the experimental methods with a more logical flow of all the processes. It is very important for the reader to know your replicates that then go into your statistics. Below I highlight some instances where the materials and methods is unclear and some suggestions. Please apply these also to the rest of the section in case I have missed something:

[Response] Thank you very much for your helpful comments and suggestions. We fully agree that clarity and transparency in the Materials and Methods section are essential. In the revised manuscript, we carefully improved its logical flow and added detailed descriptions of the sample numbers, replicates, and analytical procedures at each step. We have also clearly specified how these replicates were incorporated into the statistical analyses to enhance reproducibility and understanding.

-Line 148-158: I would even consider moving this entire paragraph to supplementary information. It makes the materials and methods larger without adding much to it and people can always download your SI if they want to read it.

[Response] Thank you for the suggestion. We have removed this paragraph to the **Supplementary Information Appendix S1**.

-Line 159-165: I would start with this paragraph in your materials and methods.

[Response] Thank you. We now start this section with this paragraph.

-Line 164: How many circular subplots?

[Response] Thank you for the comment, and we apologize for the lack of clarity. In fact, the number of subplots within each sampling plot was not fixed, as we employed a variable-area method to sample tree diversity. This approach ensured that a minimum of 100 individual trees (≥ 1 cm DBH) were recorded per plot. Given differences in tree density across plots, this method resulted in 5 to 15 circular subplots per plot. The subplots were positioned consecutively along a straight transect line within each plot. This layout minimizes potential sampling bias between larger and smaller trees and provides a representative assessment of local tree diversity and composition. We have now clarified this in the revised Methods section in lines 187–190 and added the detailed plot information in **Supporting Information Table S1**. It now reads:

"The number of subplots per sampling plot varied from 5 to 15, as we used a variable-area method (Dayananda et al., 2017). This approach, with the number of subplots adjusted according to tree density, ensured that at least 100 trees (diameter at breast height ≥ 1 cm) were recorded in each plot (refer to Table S1 and Lu et al., 2024 for plot details)."

Supporting Information Table S1. Summary of the 30 sampling plots within 17 fragmented forest patches in Xishuangbanna, Yunnan Province, China. Information includes the number of subplots, forest type, patch size, and edge distance for each plot.

Fragmented Patch	Plot	No. subplot	Forest type	Patch size (ha)	Edge distance (m)
1	1	12	Lowland	1.71	10
2	2	6	Montane	4.86	10
3	3	8	Lowland	6.3	20
4	4	8	Montane	20.7	10
5	5	7	Montane	54.9	309.8
6	6	15	Lowland	65.16	200
	7	6	Lowland	65.16	30
	8	8	Lowland	65.16	30
7	9	6	Lowland	76.14	124
8	10	6	Montane	87.57	20

9	11	8	Lowland	100.11	100
	12	6	Montane	100.11	20
10	13	7	Limestone	167.85	50
11	14	7	Lowland	329.85	517.4
	15	6	Montane	329.85	313
12	16	9	Limestone	518.31	200
	17	5	Limestone	518.31	342.3
13	18	9	Montane	925.83	50
14	19	6	Limestone	995.4	50
	20	8	Limestone	995.4	300
15	21	7	Montane	1748.25	100
16	22	7	Lowland	3283.02	469.6
	23	11	Limestone	3283.02	200
	24	8	Limestone	3283.02	485
17	25	8	Lowland	13872.87	1047
	26	5	Montane	13872.87	427
	27	7	Montane	13872.87	935
	28	10	Limestone	13872.87	254
	29	14	Lowland	13872.87	10
	30	7	Montane	13872.87	185.6

-Line 167-168: This here is why its important to mention before how many subplots were in each plot. Because now I am wondering, how many did you collect? Was it 5 per subplot? If there were 5 subplots was it then 25 per forest? Etc etc. Be detailed about it.

[Response] Thank you for raising this important point. We realize this was not clearly explained in the original submission. To clarify: the number of subplots within each plot was determined by tree density for vegetation surveys using a variable-area method and was mainly related to plant diversity survey. For soil sampling, five soil cores per plot were collected at equal distances radiating from the plot center, following the directions of the subplot arrangement (as shown in the revised **Supporting Information Figure S1**). This approach ensured spatially representative soil sampling within each plot while maintaining a consistent total number of soil samples across all plots. We have now made this explicit in the Methods (lines 193–196) in the revised manuscript. It now reads:

"In May 2021, we collected soil samples from all the 30 plots, using a five-point sampling method. Specifically, five soil cores (5 cm in diameter and 10 cm in depth) were collected at equal distances radiating outward from the center of each plot, following the arrangement directions of the subplots (refer to Fig. S1)"

Supporting Information Figure S1. Schematic diagram of the tree and soil sampling protocol. Circular subplots with a 5 m diameter (blue circles) were systematically arranged along a transect line within each plot, spaced 10 m apart. The number of subplots per plot varied from 5 to 15 depending on tree density, based on a variable-area sampling method designed to ensure a minimum of 100 trees (DBH ≥ 1 cm) per plot. Soil samples were collected at five evenly spaced points (soil cores with 5 cm in diameter and 10 cm in depth) radiating from the center of each plot, for fungal DNA extraction and soil physicochemical analyses.

-Line 169-171: How many grams total? 50 grams seems a lot for DNA extraction if you re using a soil kit that requires such little grams to work with. How many times was each replicated for the DNA?

[Response] Thank you for the comment. Five soil cores (5 cm in diameter, 10 cm in depth) collected from each plot together weighed approximately 1,200 g of soil. 50 g of these were stored at -20°C for DNA sequencing and potential future analysis. Of these, 0.5 g of soil was used for fungal DNA extraction, and each soil sample was extracted once for amplicon sequencing. This has now been clarified in the revised Methods section (lines 196–198 and 211–221). It now reads:

“The five cores from each plot were combined into a single sample (approximately 1,200 grams of soil) and transported to the laboratory on the same day.”

"Fungal DNA was extracted from 0.5 grams of soil using the DNeasy PowerSoil Kit (Mo Bio Laboratories, Carlsbad, CA, USA)... DNA extraction and PCR amplification were

each performed once per soil sample for fungal analysis."

-Line 175-202: This reads fine but it's a little bit abrupt following from the previous paragraph. The last thing you mention is the physiochemical properties which I expected to see right below but instead it then jumps over to DNA extractions.

I propose that you move the text with physiochemical properties here and then have the DNA extractions and sequencing details on a separate section right after.

[Response] Thank you for the helpful suggestion. We have now reorganized the content by moving the description of soil physicochemical property analyses directly after the soil sampling paragraph (lines 203–209). The DNA extraction and sequencing details have been placed in a separate section that follows this (lines 210–243), as recommended.

-Line 211: The soil properties were measured, not collected.

[Response] Thank you. Changed (line 203).

-Line 212-213: I Am not familiar with this. Can you describe this method? Typically studies use Loss on ignition. If this is another common method better to briefly describe it or provide a ref that does it.

[Response] Thank you for the comment. In our study, soil organic matter (OM) was determined using the dichromate oxidation method (Walkley & Black, 1934), which involves oxidizing soil organic carbon with potassium dichromate and sulfuric acid, followed by titration of the remaining dichromate to estimate soil OM content. This method is widely used in laboratories and is particularly suitable for soils with medium to low organic matter content, such as tropical regions. We have now clarified this in the revised manuscript in lines 204–207. It reads:

"Soil organic matter (OM) was determined by dichromate oxidation method, which involves oxidizing soil organic carbon with potassium dichromate and sulfuric acid, followed by titration of the remaining dichromate to estimate soil OM content (Walkley & Black, 1934)."

Reference

Walkley, A. & Black, I. A. (1934). An examination of the Degtjareff method for determining soil organic matter, and a proposed modification of the chromic acid titration

method. *Soil Science*, 37, 29–38.

-Line 232: Again without knowing your number of replicates from the M&M it becomes hard to assess the validity of the statistics. They seem well thought out but replicates are important to mention.

[Response] Thank you for your thoughtful comment. We appreciate your attention to the study design details and agree that clear reporting of replicates is essential for assessing the validity of statistical analyses. This study was based on a fragmentation transect survey including 30 plots across 17 patches, with the sampling design carefully accounting for different forest types and variation in fragmentation degree (i.e., patch size and edge distance). In each plot, five soil cores were collected for soil analyses, and tree diversity was surveyed using a variable-area method to record at least 100 tree individuals.

The number of replicates and sampling details have now been clearly clarified in the revised Methods section (lines 180–198) and **Supporting Information Tables S1 and Figure S1** (please refer to the previous responses). This study also builds upon a well-established experimental framework with several published studies based on this platform (see Dayananda et al., 2017; Pasion et al., 2018; Roeder et al. 2022; Lu et al., 2024). We appreciate your careful reading and helpful suggestion.

References

- Dayananda, S. K., Mammides, C., Lee, M. B., Liu, J. J., Pasion, B. O., Sreekar, R., Yasuda, M., Quan, R. C., Slik, J. W. F. & Tomlinson, K. W. *et al.* (2017). Topography and soil type are critical to understanding how bird and herpetofaunal communities persist in forest fragments of tropical China. *Biological Conservation*, 215, 107 – 115.
- Lu, Y., Zheng, S., Tomlinson, K. W. & Liu, J. (2024). Contrasting responses of plant herbivory and disease to local and landscape drivers. *Plant and Soil*, 501, 75 – 87.
- Roeder, M., Dossa, G. G., Cornelissen, J. H. C., Yang, X., Nakamura, A. & Tomlinson, K. W. (2022). Liana litter decomposes faster than tree litter in a multispecies and multisite experiment. *Journal of Ecology*, 110, 2433 – 2447.
- Pasion, B. O., Roeder, M., Liu, J., Yasuda, M., Corlett, R. T., Slik, J. F. & Tomlinson, K. W. (2018). Trees represent community composition of other plant life-forms, but not their diversity, abundance or responses to fragmentation. *Scientific Reports*, 8, 11374.

-Line 248-254: Was forest the independent variable here? Make it clear

[Response] Thank you for the question. Forest type here was the independent variable and we made it clear in line 292.

-Line 254: What kind of ANOVA? One way? two-way?

[Response] Thank you for the question. It's one-way anova and we made it clear in line 294.

-Line 255-257: What is the dependent and independent variable?

[Response] Thank you. In this analysis, forest type was treated as the independent variable, and the diversity and composition of each fungal functional group were the dependent variables. We have now revised the sentence in the Methods (lines 296–299) to explicitly reflect this. It now reads:

"To explore the relationships between environmental factors and diversity of each fungal guild, we first performed one-way ANOVA, treating forest type as the independent variable, to compare the diversity and composition (dependent variables) of each fungal functional guild across different forest types."

5.) The documentation of some results needs improvement (e.g. anova test results sometimes referred to as correlations). Below I highlight some of these:

-Line 289-290: Be careful how you phrase your results. Here you are referring to your graph in supplementary information which is a boxplot showing your anova results. This is no longer spearman correlations and should not be referred to as such.

[Response] Thank you for your helpful comment. We have now revised the sentence to: *"However, patch- and landscape-scale factors did not differ significantly among forest types (Fig. S5)."* in lines 348–349.

-Line 292: Again this is an anova right? This sounds like a correlation. Maybe its better to say that forest type significantly affected/shaped fungal diversity?

[Response] Sorry for the ambiguity. These results are based on ANOVA tests. We have now revised the sentence to:

"Forest type significantly affected the diversity of nine fungal functional guilds." in line 351.

In addition, we have carefully reviewed the manuscript and revised the relevant descriptions to ensure that all statistical results are accurately reported.

-Line 299-302: I might be mistaken but when you talk about composition and dissimilarity isnt it then better to show this via a PCoA? If you choose a boxplot maybe its better to describe it differently.

[Response] We agree that visualizing community composition using the full PCoA ordination is valid and informative. However, in our study, to ensure consistency with the visualization of alpha diversity metrics, we extracted the scores along the first axis of a PCoA based on Bray–Curtis dissimilarity. This is also commonly used to reduce multivariate community composition into a single quantitative variable for statistical comparison (Anthony et al., 2022; van Rijssel et al., 2022). To avoid ambiguity, the revised manuscript (lines 358–362) now explicitly describes community composition as "represented by the first axis of a PCoA (PCoA1) based on Bray-Curtis dissimilarity" and clarifies that we compared PCoA1 scores among forest types to visualize compositional differences. It now reads:

"The community composition, represented by the first axis of a PCoA (PCoA1) based on Bray-Curtis dissimilarity, of different fungal functional guilds also showed the similar pattern, with limestone forests having the greatest compositional separation, followed by lowland and mountain forests, except that soil and unspecified saprotroph showed the contrast pattern (Fig. 2c)."

References

Anthony, M. A., Crowther, T. W., Van Der Linde, S., Suz, L. M., Bidartondo, M. I., Cox, F., Schaub, M., Rautio, P., Ferretti, M. & Vesterdal, L., *et al.* (2022). Forest tree growth is linked to mycorrhizal fungal composition and function across Europe. *The ISME journal*, 16, 1327 – 1336.

van Rijssel, S. Q., Veen, G. F., Koorneef, G. J., Bakx-Schotman, J. M. T., Ten Hooven, F. C., Geisen, S. & van Der Putten, W. H. (2022). Soil microbial diversity and community composition during conversion from conventional to organic agriculture. *Molecular Ecology*, 31, 4017 – 4030.

Other comments:

6.) Line 57: What are soil carbon resources? Do you mean soil carbon in general? Then maybe its better to not call it carbon resources but just soil carbon

[Response] Thank you for the helpful suggestion. We agree that "soil carbon" is more accurate and widely used in this context. We have revised the sentence accordingly (lines 58–60) and it now reads:

"Soil saprotrophic fungi, for example, can decompose organic matter and plant litter, contributing to soil carbon accumulation and supporting plant growth (Talbot et al., 2013)."

7.) Line 58: Maybe its worth to switch Mycorrhizal to "Beneficial symbiotic fungi" to keep it more broad at an introduction level.

[Response] Thank you for the helpful suggestion. We have now changed the expression to "Beneficial symbiotic fungi can..." in line 60.

8.) Line 70: Why biotic or abiotic? Dont both interact? I would then say biotic AND abiotic

[Response] Thank you for the helpful suggestion. We have now replaced the "biotic or abiotic" with "biotic and abiotic" in line 75.

9.) Line 75-76: Is that always the case? I think it is also good to address older literature that says the opposite (acidic pH favoring fungi) as well like:

-Fungal biomass development in a chronosequence of land abandonment - van der Wal et al., 2005

-Site and management effects on soil microbial properties of subalpine meadows: a study of land abandonment along a north-south gradient in the European Alps - Zellet et al., 2001

[Response] Thank you for the valuable comment and the references. We totally agree that the relationship between soil pH and fungal diversity is not universally consistent across different environmental contexts. While the cited meta-analysis highlights a general trend of higher fungal diversity under higher pH conditions, earlier studies have also reported the opposite pattern, with acidic conditions favoring fungal communities, particularly in

forested or abandoned land systems as recommended (Zeller et al., 2001; van der Wal et al., 2005). Therefore, we revised the manuscript (lines 80–84) to incorporate this broader perspective. It now reads:

"A global meta-analysis reveals that higher soil pH is positively correlated with soil fungal richness and Shannon diversity (Zhou et al., 2020), while some regional studies have reported that acidic soils may also promote fungal biomass or growth, particularly in abandoned land systems (Zeller et al., 2001; van der Wal et al., 2006)."

10.) Line 77: I would abbreviate nitrogen to N after the first time you mention it and across the rest of the document. Similarly for other nutrients/metals etc

[Response] Thank you for the suggestion. We have now abbreviated the soil nutrients/metals the first time we mentioned. Among them, nitrogen and phosphorus first appear in line 62, soil organic matter (OM) in line 204, potassium (K), and calcium (Ca) in line 208.

11.) Line 87-88: Is there a reference to support this?

[Response] Thank you for the suggestion. Yes, we have added two related papers for references. It now reads:

"Habitat fragmentation caused by anthropogenic land use change is affecting soil fungal diversity at larger spatial scales (Le Provost et al., 2021; Raimbault et al., 2024)." in lines 100–101.

References

Le Provost, G., Thiele, J., Westphal, C., Penone, C., Allan, E., Neyret, M. van der Plas, F., Ayasse, M., Bardgett, R. D. & Birkhofer, K., et al. (2021). Contrasting responses of above-and belowground diversity to multiple components of land-use intensity. *Nature Communications*, 12, 1–13.

Raimbault, A., Brin, A., Manzi, S., Savoie, J. M., Gandois, L., Oliva, P., Fogliani, O., Roy-Camille, C., Gratacap, C. & Roy, M. (2024). Influence of habitat fragmentation and habitat amount on soil fungi communities in ancient forests. *Landscape Ecology*, 39, 19.

12.) Line 88-89: Give maybe a small definition of edge effect in this context in a

parenthesis to make this more accessible

[Response] Thank you for the valuable suggestion. We have now added a brief definition of the "edge effect" in this context (lines 102–103) to improve clarity and it now reads:

"At patch scale, fragmentation reduces patch area and increases edge effects (i.e., changes in community structures near habitat boundaries)..."

13.) Line 117: Which mixed effects? If you are mentioned THE mixed effects I would expect that the previous sentence mentions some of them but the previous sentence just mentions strong effects with no indication of them being mixed.

[Response] Thank you for the valuable comment. We agree that "mixed effects" may be unclear here. We have revised the sentence to clarify that the lack of distinction between patch- and landscape-scale factors in previous studies may obscure their individual contributions, rather than referring to any specific set of mixed effects. It now reads:

"However, most current studies do not distinguish the effects between patch- and landscape-scale environmental factors when detecting the impact of fragmentation on biodiversity, which may obscure their respective contributions (Su et al., 2022; Lu et al., 2024)." in lines 128–131.

14.) Line 213: For these two (N and P) as I mentioned you can already abbreviate them earlier in the text when you first mention them

[Response] Yes, we have already abbreviated them in line 62, so here we used the abbreviations.

15.) Line 364-377: This needs to be condensed a bit. It's a full paragraph of just summarizing results that we just read. Its good to start the discussion with a brief results summary but this is a bit too long.

[Response] Thank you for the helpful suggestion. We have now condensed this paragraph (lines 433–445) to provide a more concise and focused summary of the key findings. It reads:

"In the context of global habitat fragmentation, our study comprehensively evaluated the effects of local-, patch- and landscape-scale environmental factors on the diversity of soil

fungus functional guilds in tropical forests of Yunnan, China. We found that local tree composition was the dominant driver of the four diversity indices across the nine functional guilds, explaining on average 45.72% of the variation. While patch- and landscape-scale factors played relatively smaller roles, the landscape factor of mean patch area still significantly reduced diversity in most fungus functional guilds. Additionally, different fungus guilds showed contrasting responses: ectomycorrhizal fungus were more diverse and abundant in limestone forests in contrast to other guilds, and soil/unspecified saprotrophic fungus exhibited opposite compositional patterns in relation to tree composition. These findings highlight the distinct responses of fungus functional guilds and diversity indices to fragmentation, shaped by the pronounced ecological heterogeneity of tropical forests."

16.) Line 406-407: If I m not mistaken, in the intro you said that about basic soils and I pointed out about acidic so you have to be consistent.

[Response] Thank you for your valuable comment. As noted in the Introduction, the cited meta-analysis (line 82) emphasizes relative increases in pH as drivers of fungus diversity, rather than absolute pH categories (acidic vs. alkaline). Actually, our results largely support the positive relationship between soil pH and fungus diversity reported by Zhou et al. (2020). Although some fungus can tolerate acidic soils, we found that fungus diversity generally increased from more acidic (pH ~4) to near-neutral (pH ~7) soils across our sites, suggesting that more acidic conditions are less favorable in our system. In order to avoid confusion, we have revised the text in lines 474–476. It now reads:

"The soil pH value in our fields ranges from 4 to 7, and fungus diversity generally increased along this gradient. This suggests that while some fungus can tolerate more acidic soils, diversity tends to be lower under such conditions (Zhou et al., 2020)."

17.) Line 440-441: Cite some of those previous studies here

[Response] Thank you. We added two previous studies for reference here and it now reads:

"While previous studies often emphasize the negative effects of fragmentation (Liu et al., 2019; Yuan et al., 2024)..." in lines 509–510.

References

Liu, J. J., Coomes, D. A., Gibson, L., Hu, G., Liu, J. L., Luo, Y. Q., Wu, C. P. & Yu, M. J. (2019). Forest fragmentation in China and its effect on biodiversity. *Biological Reviews*, 94, 1636–1657.

Yuan, R., Zhang, N. & Zhang, Q. (2024). The impact of habitat loss and fragmentation on biodiversity in global protected areas. *Science of the Total Environment*, 931, 173004.

Reviewer #2 (Remarks to the Author):

Dear Authors,

I did enjoy Reading your manuscript entitled “Cross-scale drivers of soil fungal diversity in fragmented forests”, moreover exploring this question in the region of Yunnan. The manuscript is clear, rich, well illustrated and explores the importance of distinct factors across scales.

[Response] Thank you very much for your positive feedback. We appreciate your recognition of the novelty and clarity of our manuscript.

I have a few remarks – and it will be a nice example for future research on soil fungi and their ecology. It fits the scope of Communications Biology, represents an inspiring and importance peace of work. I would recommend to better contextualize the conclusions notably as you mentioned the tropical context - where landscape ecology of fungi is poorly studied. On this aspect, compared to most studies you mention from fragmented forests, your habitats are much more contrasted, and it could be mentioned.

[Response] Thank you again for encouraging us to make this aspect more explicit. Based on your suggestion, we have revised the Discussion section (first paragraph, lines 433–445; and final paragraph, lines 574–580) to better contextualize our findings in the tropical context. We now emphasize that, unlike many previous studies conducted in relatively homogeneous fragmented forests, our study spans ecologically contrasting tropical forest types in Yunnan. This provides a unique opportunity to understand soil fungal diversity responses across strong environmental gradients in a poorly studied biogeographic region. We believe this enhances the broader relevance and novelty of our study. It now reads:

" In the context of global habitat fragmentation, our study comprehensively evaluated the effects of local-, patch- and landscape-scale environmental factors on the diversity of soil fungal functional guilds in tropical forests of Yunnan, China. We found that ...These findings highlight the distinct responses of fungal functional guilds and diversity indices to fragmentation, shaped by the pronounced ecological heterogeneity of tropical forests."

"In conclusion, our results highlight the primary role of local tree composition in shaping

the diversity of dominant fungal functional guilds in fragmented tropical forests of Yunnan, with patch- and landscape-scale factors having supplementary effects. By capturing strong environmental contrasts across these fragmented tropical systems, our study helps fill a key gap in tropical fungal landscape ecology, an area that remains underrepresented in global biodiversity research. These insights offer several implications for soil fungal diversity prediction and conservation. First..."

Indeed, the differences in forest type - that could be considered as distinct habitats from a mycological point of view - shall be better introduced, as a contrasted factor to compare the importance of landscape factors.

[Response] Thank you very much for your insightful advice. We fully agree that differences in forest types represent ecologically distinct habitats and should be introduced more clearly. In the revised Introduction (lines 138–151), we added an illustration of the three forest types in our study: limestone, lowland, and montane forests. These forest types differ significantly in plant species composition, elevation ranges, soil properties, and microclimatic conditions, and therefore represent contrasting fungal habitats. It now reads:

"Furthermore, the forest type also shapes plant composition and soil conditions, impacting soil fungal diversity (Shi et al., 2014). Specifically, lowland forests (600-900 m elevation) are warmer and characterized by tall megaphanerophytes (often > 30 m) and large woody lianas (Zhu, 2006). Montane forests (above 900 m elevation), with cooler temperatures, are dominated by shorter trees with leathery, microphyllous leaves (Zhu, 2006). The diverse vegetation composition may result in more fungal diversity in lowland and montane forests (Shi et al., 2014). In contrast, limestone forests are species-poor, dominated by calciphilous species, and experience more variable microclimates, implying the presence of functionally distinct taxa of fungi (Lu et al., 2024). Moreover, soils in limestone forests are calcium-rich and near-neutral (pH ~6.5), while lowland and montane forests develop on more acidic soils (pH 4.5-5.5) (Zhu, 2006). These sharp ecological contrasts offer a unique setting for exploring cross-scale drivers of fungal diversity (Shi et al., 2014; Lu et al., 2024)."

You may also discuss some limits - especially when discussing the response of some low represented guilds (while all fungi response was not all included). Perhaps you could

also strengthen the importance of considering the impact of fragmentation not only on species richness but also other indicators, as it's also one original aspect of this study, and it extends beyond fungi ecology?

[Response] Thank you for your insightful suggestion. We have now discussed the limited representation of certain fungal guilds which constrains our ability to assess their responses in the Discussion. It now reads:

" In addition, some guilds such as AM fungi, dung and pollen saprotrophs, lichen and protistan parasites, and root endophytes, were detected at very low sequencing abundance and thus excluded from statistical analyses. Other guilds, including mycoparasites and leaf endophytes were retained in the analyses but still exhibited relatively low abundance. These results may partly reflect true ecological rarity, but also suggest potential detection biases of eDNA-based metabarcoding. Future studies should consider targeted sampling, optimized primers, or deeper sequencing to improve the detection of these guilds and better capture the functional diversity of soil fungi in tropical forests (Nilsson et al., 2019)." in lines 565–573.

Additionally, we have revised the Discussion to emphasize the importance of calculating multi-dimensional diversity indices of fungal groups to comprehensively understand their response to fragmentation, which is also applicable to broader biodiversity prediction. It now reads:

"By integrating multi-dimensional diversity indices, our study provides a comprehensive understanding of how soil fungal diversity responds to environmental factors and demonstrates distinct patterns among functional guilds." in lines 523–525.

"Finally, conservation efforts should consider the distinct responses of different fungal functional guilds, as captured through multi-dimensional diversity indices that reflect their ecological roles. This framework is applicable to broader biodiversity prediction across complex ecosystems." in lines 584–588.

More punctual comments:

First, the forest habitat you choose is particularly contrasted, with these three forest types

that could be considered as distinct habitats based on soils and tree composition, especially from a mycological point of view. Your figures 1 and S3 illustrate well these contrasts, and so do the results. It would be important to clarify it for the readers in the introduction, and in the analysis (whether forest type was included in the models or not). Please see line per line suggestion where to detail this.

[Response] Thank you very much for highlighting the importance of clarifying the ecological contrasts among the three forest types. As we have replied before, we revised the Introduction (lines 138–151) to better contextualize these differences and clearly describe the ecological heterogeneity of the studied forest types. Furthermore, we have clarified throughout the Methods and Results sections whether and how forest type was included in the statistical models, based on your line-by-line suggestions.

In the MM, you mention the description of tree composition, but in the discussion name some AM species – not ectomycorrhizal (ECM) hosts. However, given the high abundance of ECM in your dataset, why not naming these species or including a factor relative to ECM host species number? You have already done many analyses – I do not ask to make more, but at least mention it in the MM and the discussion.

[Response] We sincerely thank you for this insightful and constructive suggestion, which we fully agree is highly meaningful. In the revised Methods section (lines 252–255), we matched all recorded tree species to mycorrhizal types based on the FungalRoot v2.0 database (Soudzilovskaia et al., 2020) and recorded the presence of ECM host species in each plot (**Supporting Information Table S2**). We found that the tropical forests of Xishuangbanna are dominated by AM plants, whereas the relatively dominant species in montane forests (e.g., *Castanopsis* spp.) are ectomycorrhizal (ECM) hosts, accounting for 21.5% of trees in this forest type. This likely explains the highest ECM fungal abundance observed in montane forests, as now noted in the Discussion (lines 528–532).

Although this is an important ecological factor, we did not include ECM host proportion as a formal predictor because some species could not be confidently assigned to a mycorrhizal type or their classification remains disputed (the ECM hosts we reported here are carefully checked). Nevertheless, the approximate proportion can still serve as an informative reference. Future studies could examine the diversity of AM and/or ECM fungi at the tree species level in greater depth, using more targeted sampling and sequencing approaches to investigate their distinct responses to fragmentation in tropical forests. The revised text now reads:

"In addition, we determined the mycorrhizal type of all recorded plant species using the FungalRoot v2.0 database (Soudzilovskaia et al., 2020), and recorded the presence of ectomycorrhizal (ECM) host species in each plot (Table S2)." in Methods in lines 252–255.

" The main explanation might be that montane forests harbored, on average, 21.5% ECM-associated host plants (Table S2), dominated by *Castanopsis fleuryi* and *C. echinocarpa*, along with *Engelhardtia roxburghiana* and *E. serrata*; *Lithocarpus fohaiensis* and *L. fenestratus* were also more abundant than in other forest types (Soudzilovskaia et al., 2020)." in Discussion in lines 528–532.

Supporting Information Table S2. Richness and relative abundance of ectomycorrhizal (ECM) host tree species in each plot across three forest types (Lowland, Montane, and Limestone) in Xishuangbanna.

Plot	Forest type	ECM host richness	ECM host relative abundance
1	Lowland	0	0.00
2	Montane	4	0.17
3	Lowland	0	0.00
4	Montane	4	0.19
5	Montane	6	0.24
6	Lowland	2	0.02
7	Lowland	4	0.08
8	Lowland	0	0.00
9	Lowland	0	0.00
10	Montane	4	0.10
11	Lowland	2	0.07
12	Montane	3	0.21
13	Limestone	0	0.00
14	Lowland	1	0.05
15	Montane	4	0.33
16	Limestone	0	0.00
17	Limestone	0	0.00
18	Montane	3	0.04
19	Limestone	0	0.00
20	Limestone	0	0.00

21	Montane	4	0.14
22	Lowland	4	0.09
23	Limestone	1	0.01
24	Limestone	1	0.01
25	Lowland	1	0.02
26	Montane	3	0.32
27	Montane	3	0.25
28	Limestone	0	0.00
29	Lowland	0	0.00
30	Montane	4	0.35

References

Soudzilovskaia, N. A., Vaessen, S., Barcelo, M., He, J., Rahimlou, S., Abarenkov, K., Brundrett, M. C., Gomes, S. I. F., Merckx, V. & Tedersoo, L. (2020). FungalRoot: global online database of plant mycorrhizal associations. *New Phytologist*, 227, 955–966.

In the MM - for molecular biology part, negative controls of extraction, PCR, sequencing are not mentioned; nor the number of PCR as you mentioned pooling some products. I would recommend describing in the results a bit more this dataset, it's classical with metabarcoding data and would also reflect what are these fungal communities in tropical fragmented forests. For example, did you have a high heterogeneity of abundances before rarefying? How many sequences did you lose during the bioinformatic analyses? Based on rarefaction, you keep only half of your dataset, which is not trivial. It would be important to mention it at least.

[Response] Thank you very much for these helpful suggestions. Now, we have revised the Materials and Methods section to clearly specify the number of PCR replicates per sample and the use of negative controls for both DNA extraction and PCR. We also added the raw number of sequenced. It now reads:

"DNA extraction and PCR amplification were each performed once per soil sample for fungal analysis. Negative controls were included for both steps to monitor potential contamination, and no detectable amplification was observed by gel electrophoresis." in lines 220–223.

"In total, 2,941,702 raw sequences were generated from all samples, and were retained

for downstream analysis." in lines 227–228.

Furthermore, we added to the Results section (lines 327–340) a brief description of the dataset, including the number of OTUs identified and the variation in raw sequencing depth across all 30 samples prior to rarefaction. We also reported the sequence counts and OTU numbers of major fungal functional groups after rarefaction and functional assignment. Rarefying to the minimum sample size (23,991 sequences) inevitably resulted in some data loss; however, this commonly used approach is justified as the rarefaction curves at this depth were already approaching saturation (Supporting Information Figure S2). The supplemented text now reads:

"Among the 8,288 fungal OTUs detected, the 30 soil samples yielded an average of $39,939.9 \pm 2,163.5$ (mean \pm s.e.) sequences before rarefaction, ranging from 23,991 to 87,999 sequences per sample. After rarefaction and fungal functional guild assignment, a total of 364,654 sequences representing 2,385 OTUs were assigned to the nine fungal functional guilds, accounting for 30.4% of all fungal sequences and 28.8% of all OTUs detected (Fig. 1 and Table S5). Ectomycorrhizal fungi contributed the largest proportion of sequences (144,777) among the nine guilds, of which 89.7% belonged to the phylum Basidiomycota (Figs. 1 and S3b, Table S5). Soil saprotrophs were the second most abundant guild, comprising 90,988 sequences and exhibiting the highest OTU richness (518 OTUs) (Fig. 1 and Table S5). This guild was taxonomically diverse, including phyla of Ascomycota (26.4%), Basidiomycota (44.2%), Mortierellomycota (13.5%), and Mucoromycota (15.6%) (Fig. S3b). Foliar endophytes had the fewest sequences (2,779) and OTUs (39 OTUs) among the nine guilds, all of which belonged to phylum Ascomycota."

Based on figure S4, we can observe that all sequences were not functionally assigned which is normal in any soils. As you did not tested a model on all soil fungi – it's important to avoid conclusions on all fungi, especially as these unknown fungi represent a large part of the sequences in limestone sites. Moreover, at least in supplementary, it would be important to evaluate the representativity of each guild, in term of abundance and OTU number (a table ?). Mycoparasite, endophytes look low abundant, and it can be discussed as a limit of metabarcoding on soil eDNA. Considering that you discuss the patterns among different guild, I would suggest keeping Figure S4 in the main document.

You could have in supplementary a figure on the taxonomic diversity – to also discuss how far the level of assignment was limitant or not to distinguish the functional guilds. [Response] Thank you for these thoughtful suggestions. We agree that conclusions regarding "all fungi" should be made with caution. In the revised manuscript, we have corrected all inaccurate expressions and avoided such terminology when the analysis was limited to functionally assigned guilds. As suggested, we have added the representativity of each functional guild in terms of abundance and OTU number (see **Supporting Information Table S5**) and discussed potential limitations associated with low-abundance guilds, as well as possible approaches to improve their detection (Discussion, lines 556–573). Moreover, we have moved the original Figure S4 to the main text as Figure 1. We have also added a new **Supporting Information Figure S3** showing the taxonomic composition at the phylum level for all fungi, and expanded the discussion on how unassigned fungal phyla limited certain functional classifications, along with other possible explanations in lines 557–565. Totally, the revised Discussion now reads:

"Finally, one limitation of our study is the underrepresentation of certain fungal functional guilds. In the functional assignment, nearly 70% of OTUs could not be assigned to a specific guild, with nearly one-third unclassified at the phylum level (Fig. S3). This limitation may partly result from the exceptionally high species diversity and variability of tropical forests, where many taxa remain poorly detected or functional annotated compared with those in temperate regions (Tedersoo et al., 2020a). Such taxonomic uncertainty restricts a deeper understanding of functional patterns, and future studies could integrate multi-omics approaches or expand reference databases to improve functional resolution (Tedersoo et al., 2020a; Arıkan & Muth, 2023). In addition, some guilds such as AM fungi, dung and pollen saprotrophs, lichen and protistan parasites, and root endophytes, were detected at very low sequencing abundance and thus excluded from statistical analyses. Other guilds, including mycoparasites and leaf endophytes were retained in the analyses but still exhibited relatively low abundance. These results may partly reflect true ecological rarity, but also suggest potential detection biases of eDNA-based metabarcoding. Future studies should consider targeted sampling, optimized primers, or deeper sequencing to improve the detection of these guilds and better capture the functional diversity of soil fungi in tropical forests (Nilsson et al., 2019)."

Supporting Information Table S5. The number of sequences and OTUs of each

classified fungal functional group.

	Number of sequences	Number of OTUs
Animal parasite	13199	168
Ectomycorrhizal	144777	263
Foliar endophyte	2779	39
Litter saprotroph	25497	379
Mycoparasite	15165	77
Plant pathogen	28258	323
Soil saprotroph	90988	518
Unspecified saprotroph	22059	243
Wood saprotroph	21932	375

Supporting Information Figure S3. Soil fungal abundance based on rarefied sequence counts, assigned to the represented phyla of all fungi (a), and represented fungal functional guilds (b) from 30 sampling plots under three forest types.

Still in the methods, I was wondering how distance was taken into account, if spatial autocorrelation was significant or not? It's a basic test you could add – to be sure it's not significant, and that your closely related plots are independent. If you do not want to include it as you removed correlation later, please justify this approach.

[Response] We appreciate the valuable suggestion. Actually, we have conducted a spatial autocorrelation test (by Moran's I) in our analysis and found non-significant effects (lines 315–317). Therefore, we did not include distance as an additional factor in subsequent models. We have now revised the original text to clearly exhibit this and added the results to the **Supporting Information Table S4**. The text now reads:

"Spatial autocorrelation was assessed by Moran's I function (based on the 9 nearest neighbors) using the spdep package (V1.3–10) (Bivand & Wong, 2018), with no significant results in all models (Table S4)."

Supporting Information Table S4. Spatial autocorrelation of predicted soil fungal diversity based on multivariate linear mixed-effects models (LMM). Moran's I values and corresponding p-values are reported for the richness, Shannon diversity, abundance, and community composition of the nine fungal functional guilds.

Diversity indices	Moran's I	p-value
Animal parasite richness	-0.098	0.840
Ectomycorrhizal richness	-0.037	0.506
Foliar endophyte richness	-0.094	0.824
Litter saprotroph richness	-0.053	0.609
Mycoparasite richness	-0.077	0.743
Plant pathogen richness	-0.047	0.572
Soil saprotroph richness	-0.092	0.821
Unspecified saprotroph richness	-0.122	0.916
Wood saprotroph richness	-0.106	0.871
Animal parasite Shannon	-0.046	0.566
Ectomycorrhizal Shannon	-0.091	0.809
Foliar endophyte Shannon	-0.076	0.743
Litter saprotroph Shannon	-0.098	0.840
Mycoparasite Shannon	-0.070	0.713
Plant pathogen Shannon	-0.082	0.771
Soil saprotroph Shannon	-0.088	0.799
Unspecified saprotroph Shannon	-0.062	0.685
Wood saprotroph Shannon	-0.046	0.566
Animal parasite abundance	-0.089	0.804

Ectomycorrhizal abundance	-0.101	0.856
Foliar endophyte abundance	-0.082	0.777
Litter saprotroph abundance	0.009	0.232
Mycoparasite abundance	-0.081	0.780
Plant pathogen abundance	-0.054	0.619
Soil saprotroph abundance	-0.102	0.857
Unspecified saprotroph abundance	-0.096	0.832
Wood saprotroph abundance	-0.062	0.663
Animal parasite composition	-0.081	0.762
Ectomycorrhizal composition	-0.080	0.759
Foliar endophyte composition	-0.101	0.860
Litter saprotroph composition	-0.108	0.884
Mycoparasite composition	-0.099	0.844
Plant pathogen composition	-0.052	0.606
Soil saprotroph composition	-0.112	0.891
Unspecified saprotroph composition	-0.073	0.724
Wood saprotroph composition	-0.100	0.858

Finally, the impact of fragmentation is often evaluated on alpha diversity changes – loss of species, increase or decrease in diversity. Interestingly, you also evaluated changes in composition – and found several relationships with patch size notably. This is not really introduced while it’s an important result of your study, especially as you chose to put Figure 3 in principal and not the S7.

[Response] Thank you for the insightful comment. We agree that changes in community composition represent an important dimension of biodiversity responses to fragmentation, complementary to alpha diversity. We have now revised the Introduction to highlight the importance in incorporating community composition analyses alongside alpha diversity metrics (lines 94–99). In the Discussion, we further emphasized that different fungal guilds display distinct compositional responses to environmental factors and the potential ecological meaning (lines 548–555). It now reads:

"In addition, although many studies have primarily focused on alpha diversity metrics such as species richness of soil fungi, changes in community composition (beta diversity) can also reveal shifts in species identities and turnover (Anderson et al., 2011; Chen et al., 2017). These patterns offer equally important insights into biodiversity responses and therefore merit comprehensive and systematic investigation." in the Introduction in lines 94–99.

"In addition, the community composition of foliar endophytes, plant pathogens, and animal parasites responded more strongly to mean patch area than other fungal guilds, likely due to their closer ecological association with host species and greater sensitivity to changes in habitat size and connectivity (Grilli et al., 2017; Lu et al., 2024). These results highlight the importance of studying community composition, as shifts in species identities and turnover can have significant effects on host-microbial interactions, plant health, and ecosystem functioning (Lu et al., 2021)." in the Discussion in lines 548–555.

Additional comments :

See the main documents for line per line suggestions.

[Response] Thank you for your detailed line-by-line suggestions. We have carefully addressed each comment and incorporated the corresponding revisions in the manuscript with tracked changes.

Table S5 : Is there a significance for the colors in the table ?

[Response] Thank you for the question. Yes, the background colors in Table S5 (now Table S9) are used to distinguish different diversity indices (i.e., richness, Shannon diversity, community composition and abundance) within the response variables. We have clarified this in the table caption, which now reads: "*Background colors indicate different diversity indices of richness, Shannon diversity, community composition, and abundance.*"